# Functional 3D architecture in an intrinsically disordered E3 ligase domain facilitates ubiquitin transfer

Paul Murphy[1], Yingqi Xu[2], Sarah L. Rouse[2], Ellis G. Jaffray[1], Anna Plechanovová[1], Steve J. Matthews [2], J. Carlos Penedo [3,4] & Ronald T. Hay [1✉]

The human genome contains an estimated 600 ubiquitin E3 ligases, many of which are single-subunit E3s (ssE3s) that can bind to both substrate and ubiquitin-loaded E2 (E2~Ub). Within ssE3s structural disorder tends to be located in substrate binding and domain linking regions. RNF4 is a ssE3 ligase with a C-terminal RING domain and disordered N-terminal region containing SUMO Interactions Motifs (SIMs) required to bind SUMO modified substrates. Here we show that, although the N-terminal region of RNF4 bears no secondary structure, it maintains a compact global architecture primed for SUMO interaction. Segregated charged regions within the RNF4 N-terminus promote compaction, juxtaposing RING domain and SIMs to facilitate substrate ubiquitination. Mutations that induce a more extended shape reduce ubiquitination activity. Our result offer insight into a key step in substrate ubiquitination by a member of the largest ubiquitin ligase subtype and reveal how a defined architecture within a disordered region contributes to E3 ligase function.

[1] Centre for Gene Regulation and Expression, School of Life Sciences, University of Dundee, DD1 5EH Dundee, UK. [2] Centre for Structural Biology, Department of Life Sciences, Imperial College London, SW7 2AZ London, UK. [3] Centre of Biophotonics, School of Physics and Astronomy, University of St. Andrews, KY16 9SS St. Andrews, UK. [4] Biomedical Sciences Research Complex, School of Biology, University of St. Andrews, KY16 9ST St. Andrews, UK. ✉email: r.t.hay@dundee.ac.uk

Post-translational modifcation (PTM) of proteins with ubiquitin represents a widely used mechanism for cellular regulation. Ubiquitin is activated by an E1 enzyme, transferred to an E2 conjugating enzyme and covalently linked to substrates by one of an estimated 600 E3 ligases[1]. RING Finger Protein 4 (RNF4) is an E3 ligase that bridges the ubiquitin system with that of the Small Ubiquitin-like Modifier (SUMO) system, which it does by binding and ubiquitinating SUMO chains[2]. RNF4 and its yeast homologues have been demonstrated to function in maintaining genome integrity, requiring both its SIMs and RING domain[3–5]. RNF4 also plays a critical role in the successful treatment of acute promyelocytic leukaemia (APL). APL is driven by a chromosomal translocation resulting in a fusion protein comprised of promelocytic leukaemia protein (PML) and the retinoic acid receptor alpha (RARα). Arsenic therapy used in APL treatment leads to SUMOylation followed by RNF4-mediated degradation of PML-RARα, and clinical remission[6,7].

Approximately half of all known E3s fall into the category of ssE3s, where binding to the E2~Ub and substrate take place within the same polypeptide. Within this ssE3 subfamily, RING-type E3s dominate, with hundreds of such enzymes in the human genome[8]. Approximately 25% of all residues within ssE3s are predicted to reside in disordered regions, with these disordered regions mainly located in substrate binding domains and inter-domain linkers, while the RING domain responsible for E2 binding is ordered[9]. RNF4 follows this pattern, with its only structured content being a C-terminal RING domain required for dimerization and E2~Ub binding[10,11]. The N-termius of RNF4, representing about 75% of the whole protein, appears to lack any defined secondary structure[12]. It is within this disordered N-terminus that the four tandem SIMs of RNF4 are found, required for binding to its SUMO chain substrate. Previous studies have provided insight into how the RNF4 tandem SIMs bind to SUMO chains, and how multivalent contacts increase the binding affnity of this interaction[12–14]. While the molecular details of how the SIMs bind to the SUMO chain substrate and RING domain binds to E2~Ub have been determined, the mechanism by which SUMO chain substrate is delivered to the RING to facilitate ubiquitin transfer remains unresolved.

Considering the high number of E3 ligases in humans, there is limited information on how a bound substrate is delivered to an E3s catalytic machinery to undergo ubiquitination. However, a recent study has offered insight into RING-domain/substrate interactions that facilitate ubiquitin transfer for some of the larger multi-subunit E3s. Neddylation of Cul1, the cullin scaffold component of the CRL1$^{β\text{-TRCP}}$ complex, forms an activation module that binds to both the RING/E2~Ub and substrate receptor simultaneously. The numerous interactions involved promote ubiquitin transfer by bringing the substrate, bound to the receptor, into a proximity that allows nuceophilic attack on the E2~Ub thioester, while also promoting allosteric reactivity of the thioester bond[15]. Regarding ssE3s, the structure of RING E3 Cbl simultaneously bound to substrate peptide and E2 was assessed to define a mechanism for ubiquitin transfer[9,16]. In this structure the substrate receptor and E2 binding sites are separated by a flexible linker and the distance between target lysine and E2 active site is ~30 Å, a distance not conducive with ubiquitin transfer. Molecular dynamics simulations were applied and intramolecular diffusion deemed responsible for closing the distance between bound substrate and E2[9], although the distances obtained in the simulation were not close enough to facilitate efficient ubiquitin transfer.

Here we show that the N-terminal substrate-recognition domain of RNF4, thought to be intrinsically disordered, maintains the SIMs in a compact global architecture that facilitates SUMO binding, while an arginine-rich motif positions substrate for nucleophilic attack on RING-bound E2~Ub. Contrary to our expectation that the substrate-recognition domain of RNF4 was completely disordered, distance measurements using single-molecule Förster Resonance Energy Transfer (smFRET) and NMR paramagnetic relaxation enhancement (PRE) reveal that it adopts a defined conformation primed for SUMO interaction. Additionally, smFRET, molecular dynamics (MD) simulations and biochemical analysis indicate that electrostatic interactions involving highly charged regions linking the substrate-recognition and RING domains juxtapose those regions and mediated substrate ubiquitination. Our results reveal how a defined architecture within a disordered region contributes to substrate ubiquitination by a member of the largest family of ubiquitin E3 ligases.

## Results

**The N-terminus of RNF4 is structurally disordered**. The multi-domain protein RNF4 has a RING domain required for dimerization and ubiquitin E3 ligase activity at its C-terminus, while the N-terminal region contains the SIMs (Fig. 1a). This N-terminal region accounts for ~75% of the entire protein and displays poor chemical shift dispersion upon NMR analysis, suggesting a disordered solution state[12]. To determine if the presence of the RING domain or if RING-domain dimerization impacted the disordered state of the N-terminus, two dimensional (2D) $^1$H–$^{15}$N heteronuclear single-quantum coherence (HSQC) NMR was performed. The N-terminus alone, RNF4N (residues 32–133), was compared to RNF4 (32–194) containing the RING domain (Fig. 1b). The chemical shift pattern is near identical for the N-terminus of RNF4 in absence/presence of the RING domain, suggesting that the presence of the RING does not impose any additional order on this region, or induce a perturbation in the molecular ensemble (Fig. 1b). The high μM protein concentrations used in the NMR analysis ensure that under those conditions RNF4 is dimeric, as the $K_D$ for RING dimerization is ~180 nM[17]. Thus it can be concluded that the RNF4 N-termini remain disordered when the ligase is in its dimeric active state.

**RNF4 SIM region is compact and primed to engage SUMO chains**. Previous NMR analysis of a complex between the RNF4 N-terminal region encompassing SIMs 2/3 and diSUMO showed that the SIM2/3 region adopts a kinked conformation that restrains the two linked SUMOs[12]. We initially assumed that in the absence of SUMO chains the SIM-containing region exhibits an extended, disordered conformation that becomes more ordered and compact upon binding to SUMO chains. To probe the solution structure and dynamics of the RNF4 SIMs region, we performed smFRET measurements. A series of peptides encompassing the RNF4 N-terminus (RNF4N) with pairs of cysteine residues were generated (RNF4N 30/57, 44/70 and 57/84), and subsequently labelled with Cy3B/Alexa647 FRET pair ($R_0$~60 Å[18]). A C-terminal Avitag allowed for biotinylated peptide immobilization on a neutravidin coated surface for smFRET measurements (Supplementary Fig. 1). Dye labelling did not interfere with SUMO chain binding (Supplementary Fig. 2). Using the kinked solution structure of the SIM2/3 region bound to a SUMO dimer, obtained from NMR[12], and the Förster distance ($R_0$ 60 Å) of the FRET pair, we predicted a FRET efficiency value ($E_{FRET}$) of 0.6 using the accessible volume method[19] (Fig. 2a). As a guide, unbound stretched state modelling using a peptide scaffold (derived from PDB: 5M1U) gave an $E_{FRET}$ value of 0.23 (Fig. 2a). smFRET histograms of the RNF4N peptides 30/57, 44/70 and 57/84 showed a major population (>90%) with a high-FRET state ($E_{FRET}$ ~0.6–0.7) (Fig. 2b; Supplementary Table 1), consistent with a kinked structure even in the absence of SUMO chains. The remaining population (~10%) displayed a low-FRET value

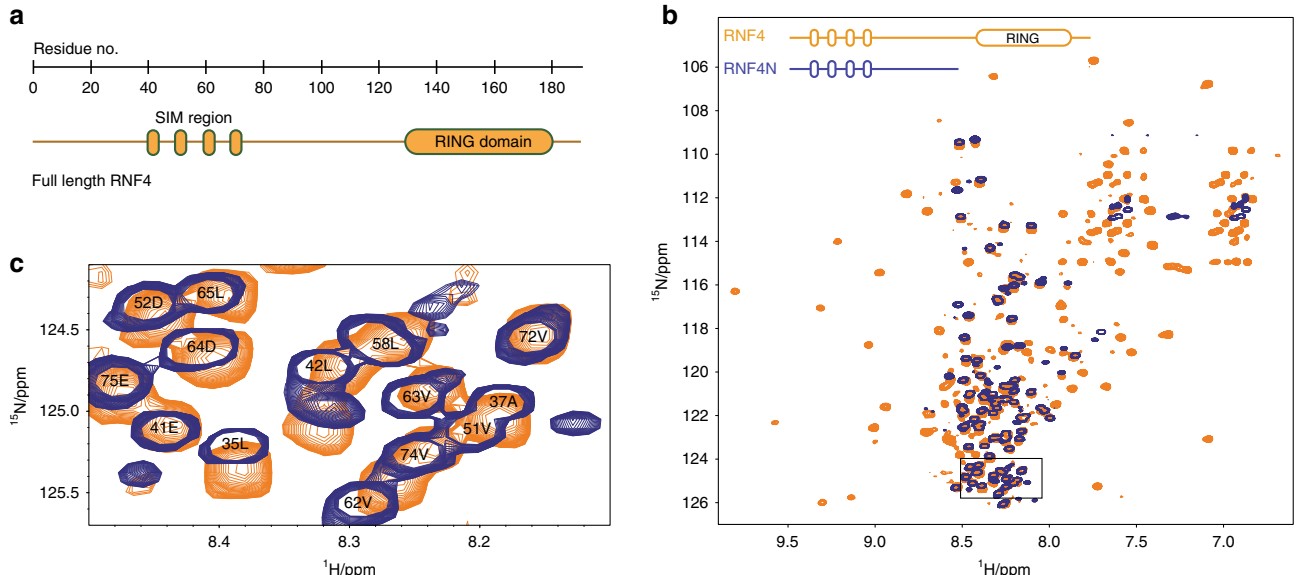

**Fig. 1 The N-terminus of RNF4 is structurally disordered. a** Domain layout of RNF4 to scale (residues 1–194). **b** Overlay of $^1$H–$^{15}$N HSQC spectra for RNF4 (32–194; yellow) and RNF4N (32–133; blue) showing near identical chemical shift patterns for the N-terminal region. Note the RNF4 spectrum was recorded with TROSY magnetisation selection and adjusted for the peak offsets. Peaks only observed in RNF4 spectrum arise from the structured C-terminal RING Domain. Near identical peak patterns for the N-terminal regions indicate that the conformational properties are the same in both samples. **c** Enlarged spectra corresponding to boxed area in B highlighting residues in the SIMs and SIMs/RING domain linker region.

($E_{FRET}$ ~0.25–0.3), similar to that predicted for the extended conformation (Fig. 2a). The addition of SUMO chains did not alter significantly the $E_{FRET}$ value ($\Delta E_{FRET} < 0.06$) (Fig. 2c) or the relative contribution of each population, suggesting the SIMs region already exists in a compact shape primed for engagement with a SUMO polymer (addition of SUMO chains ranging from 5–25 μM, $K_D$ ~2 μM). Positioning the FRET dyes outside all four SIMs, RNF4N 37/77, resulted in an $E_{FRET}$ ~0.4 (Fig. 2b). This is indicative of a compact conformation as the interdye distance in RNF4N 37/77 is such that in an extended conformation it would be beyond FRET detection. Single-molecule FRET trajectories of each RNF4N peptide displayed a single long-lived FRET state with no conformational dynamics, lasting ~10–20 seconds before photobleaching occurred (Fig. 2d; Supplementary Fig. 3–6). The long-lived FRET trajectories observed with and without SUMO were further investigated by cross-correlation analysis (Fig. 3) and we found no evidence for dynamic transitions between conformational states for any of the RNF4N peptides. We concluded that the observed FRET levels represent static conformations, at least under the conditions and within the timescale of our experiments (200 ms integration time), and that variations in the width of the FRET populations between RNF4N peptides (Supplementary Tables 1–3) represent slightly different levels of static heterogeneity within each peptide conformer.

The SIMs in RNF4 are short stretches of hydrophobic residues separated by polar residues. To investigate whether SIM–SIM interactions determined the compact shape, RNF4N ΔSIMs bearing mutations that abolish SUMO binding and reduce their hydrophobicity (Supplementary Fig. 2) were labelled identically to wild-type (WT) RNF4N. Single-molecule FRET histograms of RNF4 ΔSIMs showed ~80% of molecules in a high-FRET state (Supplementary Fig. 7A) similar to WT ($\Delta E_{FRET} < 0.03$) but with a slightly higher occupancy of the low-FRET state ($E_{FRET}$ ~0.3) (Supplementary Fig. 7B). smFRET trajectories of RNF4 ΔSIMs also showed no evidence of dynamic transitions between these states (Supplementary Fig. 8). Interactions between the hydrophobic SIMs are therefore unlikely the major driving force for the observed compaction within the SIMs region.

**RNF4 N-terminus adopts a compact shape**. To provide further insight into the global conformation of RNF4N we employed NMR paramagnetic relaxation enhancement (PRE) (Fig. 4a, c). MTSL spin labels were introduced at the N-terminus, centre (residue 70) and C-terminus of $^{15}$N-enriched RNF4N (Supplementary Fig. 9). PREs are observed by comparison of peak intensities from 2D $^1$H–$^{15}$N HSQC spectra measured on MTSL-labelled samples before and after reduction by ascorbate. Relaxation rates for residues located within ~25 Å of the spin label are enhanced and result in peaks with reduced intensity. Comparison of $^1$H–$^{15}$N HSQC spectrum of RNF4N before (Fig. 4b, upper panel) and after (Fig. 4b, lower panel) ascorbate reduction indicated a number of significant intensity changes. Using previously obtained NMR assignments[12], PREs could be mapped to sequence positions. PREs are observed for residues 74 and 75 in response to a spin label present at either end (Fig. 4c), indicating that both termini spend a significant proportion of their time within about 25 Å of these central residues. This is particularly apparent in the C-terminally MTSL-labelled RNF4N with intensity ratios (before/after MTSL reduction) of 0.75 and 0.71, respectively (Fig. 4b). Similarly, for RNF4N labelled centrally (S70C), PREs were observed at both termini (Fig. 4c). In RNF4N (S70C) the distance between the spin label and the termini would be over 100 Å in a purely extended conformation, therefore the PRE analysis suggests that the termini are folded back towards the middle of the sequence. Additional RNF4N peptides labelled at residue 44 or 58 show PREs across the whole SIM region indicating a compact shape (Supplementary Fig. 10), in-line with the smFRET analysis (Fig. 2).

Residues 74 and 75, which displayed PREs from RNF4N labelled at either termini, sit directly adjacent to a region (residues 76–89) that displayed line broadening in which signals were either absent or displayed low signal-to-noise in the HSQC spectra (Supplementary Figs. 11–15). Besides residue 82 which displayed PREs, residue 77 could be tentatively assigned but not quantified due to absence of signal in the paramagnetic spectrum. The surrounding residues 76,78–81 and 83–89 were broadened beyond detection (and thus lack assignment) in the

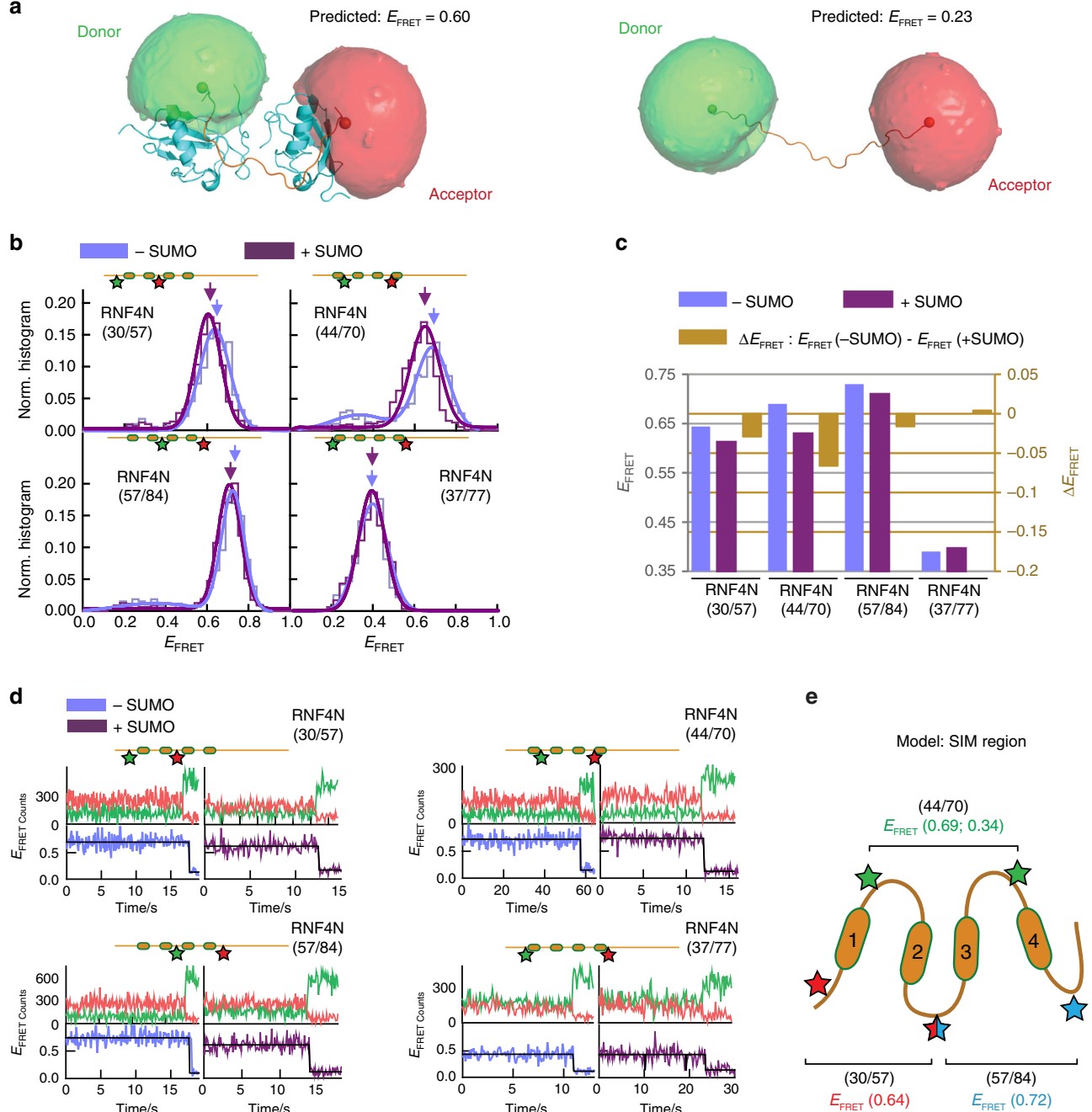

**Fig. 2 Role of RNF4 N-terminus conformation in substrate binding. a** NMR structure (PDB: 2MP2) of the SIM2/3 region of RNF4 (orange) bound to a SUMO2 dimer (cyan)[12] modified to include the donor (green) and acceptor (red) fluorophores used for single-molecule analysis of RNF4, with a predicted FRET efficiency ($E_{FRET}$) of 0.6 (left). A modelled peptide (PRB: 5M1U) with similar interdye distance but a stretched conformation produced a predicted $E_{FRET}$ value of 0.23 (right). **b** Normalised single-molecule FRET histograms of RNF4N peptides with positions of the donor (green) and acceptor (red) dyes indicated (labelled residue numbers inset). RNF4N was measured in the absence (blue) and presence (purple) of SUMO chains. Arrows indicate FRET populations used for comparison with/without SUMO (in **c**). Single-molecule histograms were built from more than 500 molecules. **c** Relative change in $E_{FRET}$ upon SUMO addition (dark yellow) from **b** was calculated as: $E_{FRET}$(+SUMO) − $E_{FRET}$(−SUMO). **d** Representative single-molecule trajectories obtained for RNF4N peptides in **b**. Top panels show donor (green) and acceptor (red) intensity signals. Bottom panels show the $E_{FRET}$ trajectory derived from the donor/acceptor intensity traces in the absence (blue) and presence (purple) of SUMO chains. **e** Model of the SIM region of RNF4 displaying a compact shape in-line with the single-molecule analysis. The position of FRET dye pairs for RNF4N peptides along with $E_{FRET}$ values corresponding to histograms in **b** are shown (30/57, red; 44/70, green; 57/84, blue). Source data are provided as a Source Data file.

non-paramagnetic spectrum and therefore the data are not available for the PRE even qualitatively. These observations are likely to be a consequence of conformational exchange for this region and may be associated with the observed compaction. Inspection of the RNF4 sequence revealed this region

encompassed an arginine-rich motif (Fig. 4a) that may contribute to electrostatic interactions within RNF4N. To validate the NMR PRE findings, smFRET RNF4N peptides were generated with FRET dyes split between the N-terminus and arginine-rich motif (residues 29/84), or the C-terminus and arginine-rich motif

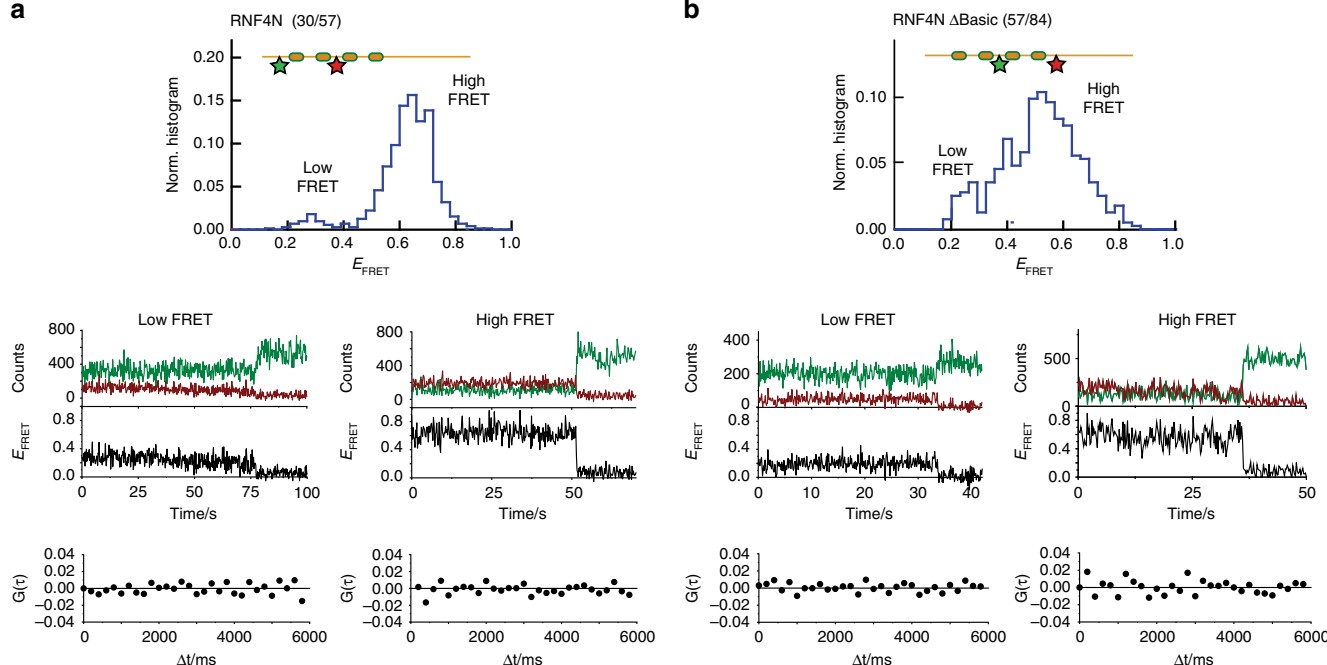

**Fig. 3 Cross-correlation analysis of RNF4N smFRET peptides. a** Single-molecule FRET histogram obtained for RNF4N 30/57 (top panel), representative low- and high-FRET trajectories (middle panel) and cross-correlation curves (bottom panels) obtained for the traces shown in the middle panels. **b** Single-molecule FRET histogram obtained for RNF4N ΔBasic 57/84 (top panel), representative low- and high-FRET trajectories (middle panel) and corresponding cross-correlation curves (bottom panel). Histogram in A from Fig. 2b, histogram in B from Supplementary Fig. 17. Source data are provided as a Source Data file.

(residues 84/119). The distance between FRET dyes in both smFRET peptides means an extended conformation would be beyond FRET detection. For both RNF4N peptides tested FRET signals were observed, RNF4N 29/84 $E_{FRET}$ ~0.4 and RNF4N 84/119 $E_{FRET}$ ~0.7, confirming that both termini are folded back towards the arginine-rich motif in the centre of the peptide (Fig. 4d).

**Highly charged regions in RNF4 promote a compact shape**. With the N-terminus of RNF4 adopting a compact shape with both termini folding back towards the arginine-rich motif in the centre, we focused our efforts on analysing the charge distribution of the RNF4 peptide. A hydropathy plot was generated to display the charge characteristics of the amino-acid side chains throughout the protein (Fig. 5a). The highly basic arginine-rich motif (residues 77–88) is flanked on the N-terminal side by the hydrophobic SIMs embedded in a highly acidic environment (residues 32–76, 13 acidic), and on the C-terminal side by a cluster of acidic residues (residues 100–109, six acidic). Analysis of RNF4 sequences across multiple species revealed strong conservation of this charge segregation (Supplementary Fig. 16). To test the potential role of electrostatic interactions in promoting the observed compact shape in RNF4, a pair of RNF4N mutants were produced for smFRET analysis (Fig. 5b). The first mutant had the basic charge removed from the arginine-rich motif (RNF4N ΔBasic), while the second had the acidic charge removed from the same linker region (RNF4N ΔAcidic). A combination of alanine and serine mutations were used, as alanine alone caused peptide aggregation. Both RNF4N mutant peptides displayed a different size exclusion chromatography profile than the WT, suggesting a change in the hydrodynamic radius of the peptide (Fig. 5c). RNF4N ΔBasic eluted earlier than WT, while RNF4N ΔAcidic eluted later, suggesting a more extended and more compact peptide, respectively. smFRET analysis revealed that

compared to the WT peptide ($E_{FRET}$ (0.68 ± 0.17)) the RNF4N ΔBasic peptide displayed a reduced $E_{FRET}$ (0.53 ± 0.1) consistent with an extended conformation, while the RNF4N ΔAcidic peptide displayed a modestly increased $E_{FRET}$ (0.72 ± 0.15) consistent with a compact conformation, (Fig. 5d). Thus the smFRET data shows the same trend in peptide conformation as the size exclusion analysis of the same peptides, although the shift between WT and ΔAcidic is less pronounced with smFRET compared with size exclusion analysis. This reduced shift observed between techniques (Fig. 5c, d) likely arises as the gel filtration analysis reports on the whole peptide, while smFRET only reports on the distance between the two dyes, in this case the arginine-rich motif to C-terminus. The observed extended conformation with RNF4N ΔBasic may arise because the lost basic charge leads to electrostatic repulsion between the acid SIM region (residues 32–76) and the acidic patch (residues 100–109) between the basic region and the RING. In contrast, loss of the acidic patch (residues 100–109) could increase the interaction between the basic arginine-rich motif and the acid SIMs, resulting in the modest compaction observed in RNF4N ΔAcidic. The arginine-rich motif resides more to the centre of RNF4N and loss of the basic charge here has the largest impact on the peptide shape. Its removal (ΔBasic) results in the whole peptide adopting a significantly more extended conformation (lower FRET) compared to WT (Supplementary Figs. 17–20).

To further investigate the role of electrostatic interactions in promoting the observed compact shape in RNF4N, molecular dynamics simulations using the SIRAH forcefield were performed[20]. Five independent simulations were carried out measuring the distance from acidic residues between SIM2/3 and the midpoint of the arginine-rich motif over time. Simulations for RNF4N WT displayed shorter distances between these peptide regions compared to RNF4N ΔBasic, consistent with a more compact shape (Fig. 5e). In addition, conformational clustering was performed where an initially fully extended model

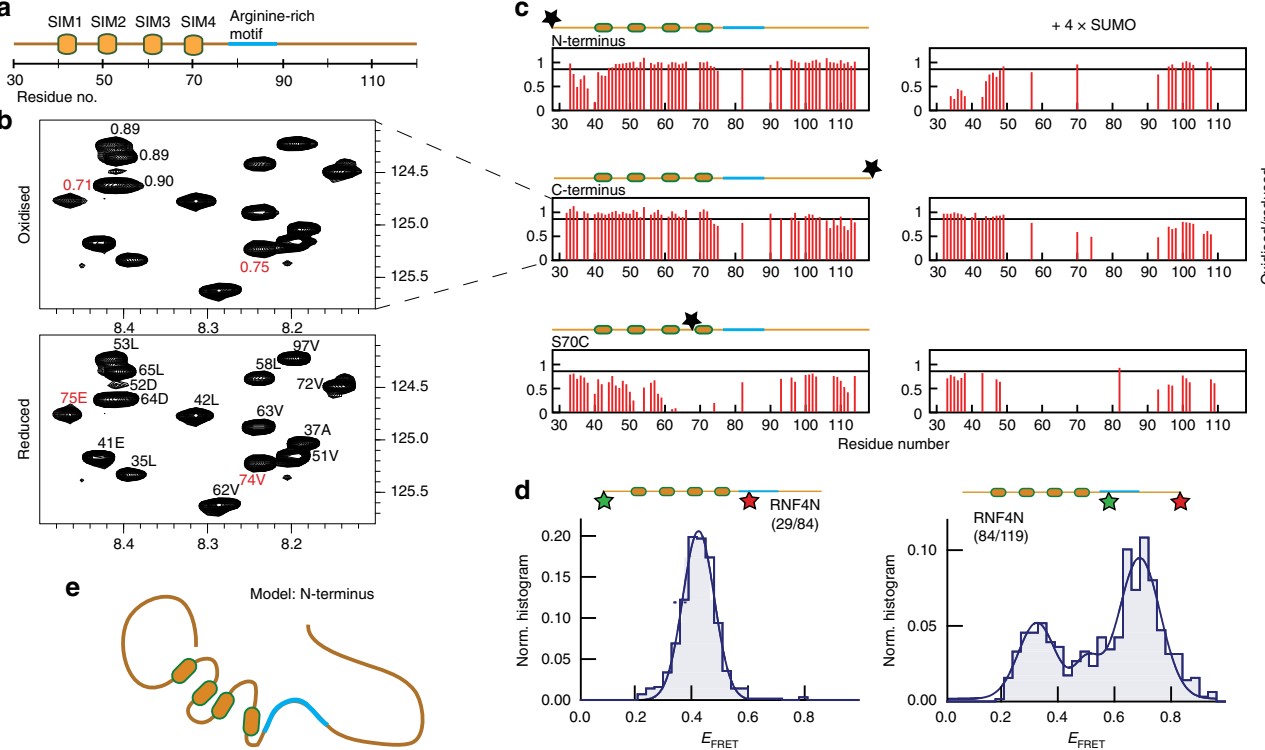

**Fig. 4 NMR and smFRET reveal compact shape of entire RNF4N-terminus. a** RNF4N peptide layout used in B and C. **b** ¹H–¹⁵N HSQC spectrum of RNF4N labelled at the C-terminus with an MTSL spin label (top). Reference 1H-15N HSQC spectrum of reduced RNF4N using peak assignment established previously[12] (bottom). Peaks with significant ratio changes adjacent to the arginine-rich motif, 74E and 75 V (red), indicated. **c** Histograms of HSQC peak intensity ratios plotted for RNF4N labelled at the termini and S70C (left) and in complex with SUMO chains (right). Peaks are absent due to severe peak broadening in the SUMO complex. Significant intensity reductions are below 0.85 (black line) (corresponding ¹H–¹⁵N HSQC spectra, Supplementary Figs. 11–13). **d** Single-molecule FRET histograms obtained from RNF4N peptides labelled with donor/acceptor FRET dyes at the N-terminus and basic region (left) or C-terminus and basic region (right) (labelled residue numbers inset). Single-molecule histograms were built from more than 400 molecules. **e** Model of RNF4N displaying both termini folding towards the centre of the peptide, and with the basic arginine-rich motif adjacent to the SIMs highlighted (blue). Source data are provided as a Source Data file.

of RNF4N WT adopted predominantly compact conformations involving loop or hairpin structures, that were mediated by interactions between the arginine-rich motif and negative charged residues within the SIM region, as well as the C-terminal acidic cluster from the SIMs/RING-domain linker. The RNF4N ΔBasic mutant displayed some reduction in loop and hairpin structures, and the conformational clustering analysis revealed models that tended towards more extended conformations (Fig. 5f and Supplementary Fig. 21). These analyses highlight how intramolecular electrostatic interactions promote the observed compact shape in the N-terminus of RNF4. It is therefore likely that in the context of the full-length RNF4 protein, the N-terminus brings the RING domain in closer proximity to the SIMs, thereby facilitating ubiquitination of the bound SUMO.

**Arginine-rich motif promotes interaction of RING and SIMs.** NMR and smFRET analysis of RNF4N revealed that this section of the RNF4 protein maintains a compact shape. To investigate if this compaction promoted interaction between the SIMs and RING domain of RNF4, full-length RNF4 was produced for smFRET analysis bearing one FRET dye in the SIMs region (residue 70), and another in the RING domain (residue 144) (Supplementary Fig. 22). Although the RING domain contains seven cysteine residues required for zinc coordination, a surface exposed serine reside (S144) could be mutated to cysteine and labelled with a maleimide functionalized dye without modifying the zinc coordinators. In addition to the WT protein, a ΔBasic

version of RNF4 was also produced for smFRET to probe the effect of the more open/extended conformation. Addition of the dye labelled RNF4 proteins into a substrate dependent in vitro ubiquitination assay revealed that the labelled proteins retained ubiquitin E3 ligase activity, thus confirming the integrity of the labelled RING domain (Supplementary Fig. 22E). smFRET histogram for RNF4 WT displays a prominent high-FRET population ($E_{FRET}$ ~0.5–0.6), highlighting a state with proximal SIMs and RING domain (Fig. 6b), along with low-FRET ($E_{FRET}$ ~0.3) population. As expected, RNF4 ΔBasic displayed a significant reduction in the high-FRET population compared to the WT protein, consistent with the more extended conformation (Fig. 6c). Single-molecule FRET trajectories of both RNF4 proteins displayed single long-lived FRET states with minimal conformation dynamics between these states lasting ~10–60 seconds prior to photobleaching (Fig. 6b, c). The addition of SUMO chains reduced the proportion of molecules in the high-FRET state for both RNF4 WT and ΔBasic, but in both the absence and presence of SUMO chains the RNF4 WT had two times as many molecules in the high-FRET state as the ΔBasic (Supplementary Fig. 23). The decrease in the number of molecules in the high-FRET state in the presence of SUMO chains may be a consequence of the high concentrations of SUMO chains (~10 μM) used in the assay.

**Loss of basic region reduces ubiquitination activity.** To address the role of the basic region in the function of RNF4 as a SUMO

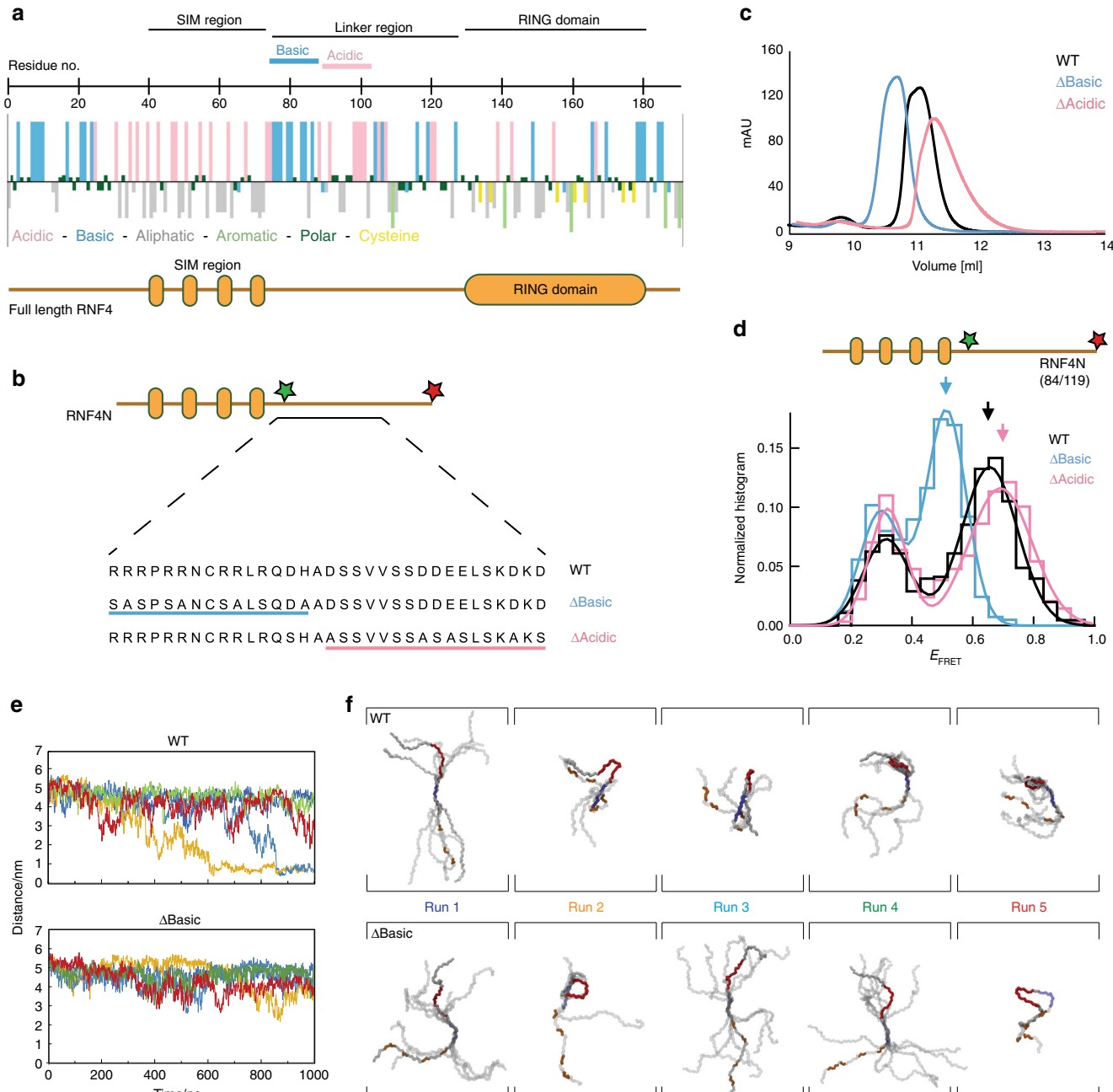

**Fig. 5 Charged regions in the RNF4 linker promote a compact but not collapsed shape. a** hydropathy plot of RNF4 revealing the charge characteristics of the amino-acid side chains. The SIMs and RING domain are highlighted, along with the charged linker that connects these two domains. **b** layout of the RNF4N peptide used for single-molecule analysis. Variants of the RNF4N peptide were produced removing either the basic or acid charge from the SIMs/ RING-domain linker region. The mutations used are displayed against the WT peptide. **c** size exclusion chromatography profile of the WT RNF4N peptide (black) along with ΔBasic (blue) and ΔAcidic (magenta) variants. **d** single-molecule FRET histograms obtained from RNF4N peptides labelled with donor/ acceptor FRET dyes at C-terminus and basic region (labelled residue numbers inset). Arrows inset highlight the FRET efficiency average for WT (black), ΔBasic (blue) and ΔAcidic (magenta) RNF4N peptides. Single-molecule histograms were built from more than 400 molecules. **e** Distance measurements over time for five independent simulations between acidic residues in the SIM2/3 linker (defined by E59) and the midpoint of the arginine-rich motif for RNF4N WT (top) and ΔBasic (bottom). **f** Conformational clustering of RNF4N peptides (WT top; ΔBasic bottom) generated from residue-residue contacts over the final 100 ns of each run trajectory from **e**. The SIMs (orange), arginine-rich region (blue) and acidic region in the SIMs/RING-domain linker (red) are highlighted. Source data are provided as a Source Data file.

targeted ubiquitin E3 ligase, the ubiquitination activity of full-length RNF4 ΔBasic was compared to WT. We monitored the response of RNF4 WT and ΔBasic to increasing 4xSUMO2 substrate concentrations by measuring the utilisation of fluorescently labelled ubiquitin in an in vitro reaction with purified E1 and UbcH5a. Reaction products were fractionated by SDS-PAGE and the

distribution of ubiquitin revealed (Fig. 7a) and quantified by in-gel fluorescence (Fig. 7b). In the presence of 4xSUMO substrate, RNF4 catalysed the efficient incorporation of free ubiquitin into higher molecular weight material and this was severely compromised with RNF4 ΔBasic (Fig. 7a, b). To compare the rates of ubiquitination, we carried out a fluorescence polarisation (FP) assay that measures in

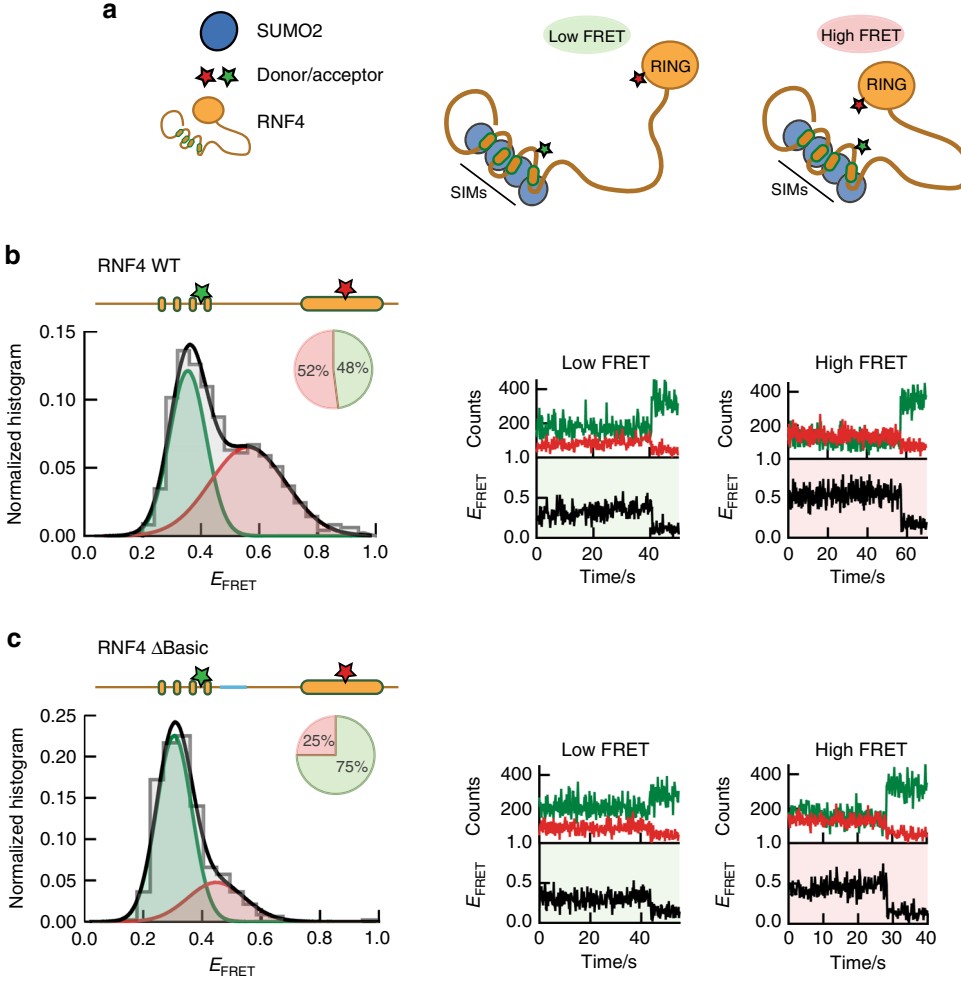

**Fig. 6 Single-molecule analysis of full-length RNF4 confirms compact shape. a** Model of full-length RNF4 displaying potential low-FRET and high-FRET states. **b** Single-molecule FRET histograms obtained from full-length RNF4 WT, labelled with donor/acceptor FRET dyes at SIMs and RING domain (residues 70,144; illustrated inset) (left). Pie charts indicate the percentage of molecules in a low-FRET (green) or high-FRET (red) state. Representative single-molecule trajectories obtained for RNF4 WT (right). **c** As in **b** but analysis performed on RNF4 ΔBasic. Single-molecule histograms were built from more than 500 molecules. Source data are provided as a Source Data file.

real-time the incorporation of fluorescently labelled ubiquitin into higher molecular weight products (Fig. 7c). Analysis of the initial rate of the reactions indicated that both RNF4 WT and ΔBasic display a lag phase before incorporation of ubiquitin accelerates. For RNF4 ΔBasic this lag phase is extended and the rate of incorporation remains slower, although the data suggests that later incorporations into growing ubiquitin chains reaches a similar rate to RNF4 WT (Fig. 7c). To directly compare the ability of RNF4 WT and ΔBasic to ubiquitinate substrate, the rate of fluorescently labelled 4×SUMO2 being ubiquitinated was determined. Reaction products were revealed (Fig. 7d) and quantified (Fig. 7e) by in-gel fluorescence. Compared to RNF4 WT, RNF4 ΔBasic had a 3-fold reduced initial rate of ubiquitination (Fig. 7e). Although mutations to the basic region in RNF4 ΔBasic are more than 40 residues from the RING domain, it was important to establish their influence on intrinsic RING activity. We therefore performed a substrate-independent lysine discharge assay that measures the ability of free lysine to act as the nucleophile and attack the thioester bond between ubiquitin and the E2, resulting in the release of free ubiquitin. In the presence of an active E3 ligase the thioester bond is activated and becomes susceptible to nucleophilic attack by lysine. The rate at which fluorescently labelled ubiquitin was released was determined by in-gel fluorescence and revealed that both RNF4 WT and ΔBasic released ubiquitin at similar rates (Fig. 7f,

g), indicating that the mutations did not affect the intrinsic catalytic activity of the RING. To rule out differences in binding of SUMO as an explanation for the differences in ubiquitination activity between RNF4 WT and ΔBasic the affinity of the two proteins for fluorescently labelled 4xSUMO2 was determined by FP (Fig. 7h). The $K_D$ for binding were 1.6 μM and 2.6 μM for RNF4 and RNF4 ΔBasic, respectively. These similar $K_D$ values are in-line with previous determinations by ITC (1.6–2.5 μM)[12,13] and the small differences are unlikely to be responsible for any differences in ubiquitination activity, particularly as assays were carried out with substrate concentrations well above the $K_D$ for SUMO chain binding. The delay in ubiquitination activity observed in RNF4 ΔBasic highlights the importance for the observed compaction in the RNF4N-terminus, as it promotes interaction between the SIMs and RING domain and facilitates substrate ubiquitination.

Biochemical analysis of RNF4 revealed the importance of the arginine-rich motif in promoting ubiquitination through juxtaposition of the RING domain with the SIMs. To determine the impact of mutating the arginine-rich motif had on RNF4 activity in a physiological setting, either RNF4 WT or RNF4 ΔBasic were stably expressed in a U2OS RNF4$^{-/-}$ cell line (Supplementary Fig. 19A, B). Under normal circumstances arsenic treatment of cells leads to PML SUMOylation which in turn recruits RNF4,

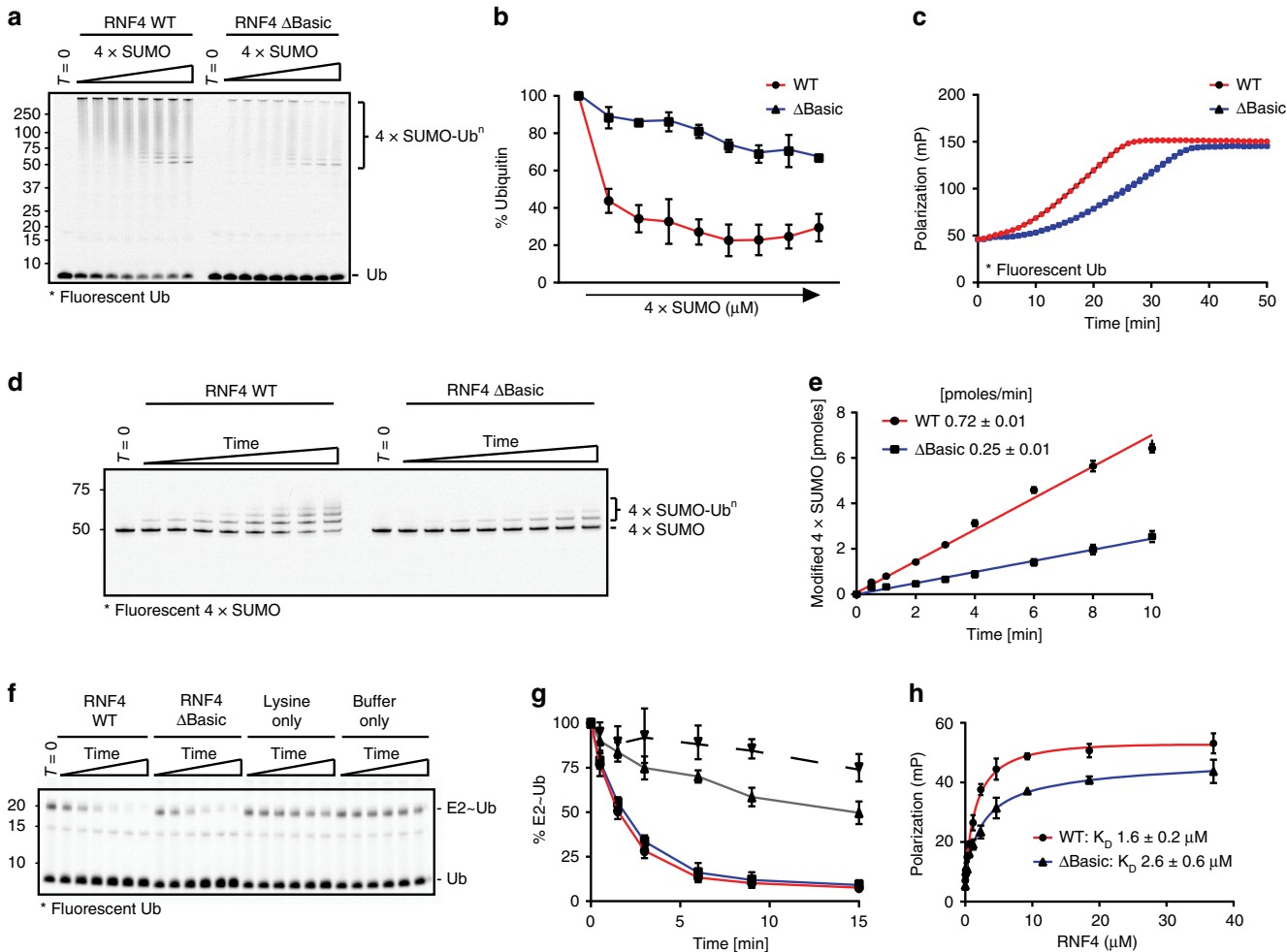

**Fig. 7 Basic region mutant displays reduced ubiquitination activity. a** Ubiquitination activity of RNF4 wild-type (WT) and ΔBasic mutant in response to increasing substrate (4xSUMO) concentrations. Assays (15 min) contained E1 activating enzyme, UbcH5a and fluorescently labelled ubiquitin. T = 0 is without ATP. Reaction products were fractionated by SDS-PAGE, followed by in-gel fluorescence. **b** Quantitation of fluorescent free ubiquitin in A (WT red, ΔBasic blue). Ubiquitin levels at T = 0 represented 100%. **c** Fluorescence polarization real-time ubiquitination assay comparing activity of WT (red) against ΔBasic (blue). Assays contained 4×SUMO, E1, UbcH5a and fluorescent ubiquitin. Reactions initiated with ATP addition and FP measured over 50 minutes. **d** Substrate ubiquitination activity assay containing fluorescently labelled 4×SUMO, ubiquitin, E1, UbcH5a and either RNF4 WT or RNF4 ΔBasic. Reaction products analysed as in **a**. **e** Quantitation of D for RNF4 WT (red) and RNF4 ΔBasic (blue). **f** Lysine discharge assays using fluorescently labelled ubiquitin. The discharge of ubiquitin from the E2 (E2~Ub) was analysed by SDS-PAGE over time in the presence of RNF4 WT, RNF4 ΔBasic, lysine only, or buffer only. **g** Measurement of fluorescence from the E2~Ub as seen in F. T = 0 represents 100% E2~Ub (WT, red; ΔBasic, blue; lysine only, grey; buffer only, dashed). **h** Binding affinity of RNF4 for 4xSUMO via FP. A fixed 4xSUMO concentration was titrated with RNF4 (WT, red; ΔBasic, blue) to saturating concentrations. For each graph (**b**, **c**, **e**, **g**, **h**) data represent mean ± standard deviation (the experiment was done in triplicate, n = 3). Experiments were carried out three times with similar results. Experiments in **a**, **d** and **f** were carried out three times with similar results. Source data are provided as a Source Data file.

followed by RNF4-mediated ubiquitination and proteasomal degradation of PML[6,7]. Thus, RNF4[−/−] cells or the same cells expressing either RNF4 WT or RNF4 ΔBasic were treated with arsenic and PML degradation was followed by immunofluorescence measurement of PML bodies. Determining the number of PML bodies present in the nucleus at 1 and 4 h post arsenic treatment, U2OS cells expressing RNF4 WT reduced PML body numbers by 32.8%, when compared to the knockout cells, while RNF4 ΔBasic reduced PML body numbers by 24.1% when compared to the knockout cells. The difference between WT and ΔBasic is significant ($p = 6.2 \times 10^{-4}$) and represents a 25% reduction in the ability of the ΔBasic to reduce PML body numbers when compared to WT (Supplementary Fig. 24C, D). Considering the important role of RNF4 in arsenic induced degradation of PML, arsenic treatment had little impact on PML body number in RNF[−/−] cells. Thus, it is likely that the reduced

ubiquitination activity of the RNF4 ΔBasic in vitro is manifested in a reduced ability to clear PML bodies in response to arsenic treatment in vivo. The observation that RNF4 WT and ΔBasic displayed significantly increased PML clearance compared with RNF4[−/−] cells is consistent with our biochemical analysis (Fig. 7) showing that RNF4 ΔBasic is a functional but impaired E3 ligase compared to RNF4 WT. The basic arginine-rich motif thus functions as one component of an E3 ligase by helping to integrate substrate binding and catalytic activity.

## Discussion

Intrinsic disorder is found in entire proteins or protein regions and has a number of common characteristics that prevent protein folding. These unstructured regions tend to be long (>30 residues), rich in charged residues while containing minimal bulky

hydrophobic residues, and usually located at borders or inter-connecting regions between globular domains[21,22]. Short linear motifs are commonly found in disordered regions and are often grouped together promoting multivalent interactions[23]. RNF4 conforms to a number of these disordered traits. The N-terminus of RNF4 (~130 residues) bears no defined secondary structure and possesses a significant number of charged residues (Fig. 5a). The four tandem SIMs are themselves short linear motifs that are composed primarily of hydrophobic residues[24,25], within an acidic background, which likely prevents inappropriate hydrophobic contacts between SIMs by electrostatic repulsion. The SIMs are essential for RNF4 recruitment and activation by SUMO chains[17]. Previous NMR analysis of a SUMO2-dimer revealed that the SUMO2 units within a chain move independently from one another[12]. Therefore, the compact but not completely collapsed arrangement of the N-terminal SIM region could function to allow optimised presentation of the SIMs to the multiple hydrophobic binding grooves present on the target SUMO chain[12,26].

Despite the N-terminus of RNF4 bearing no secondary structure[12], our smFRET and NMR data show that this region indeed adopts a compact global architecture. Intrinsically disordered regions tend to have a high net charge that causes peptide chain expansion when charges are evenly distributed across the sequence[27]. The observation that the RNF4N-terminus adopts a compact configuration likely arises from its pattern of segmented charged regions, with the location of the arginine-rich motif bookended by acidic regions promoting folding back and compaction (Fig. 5a). This is in-line with a detailed molecular dynamics simulation analysis of charge distribution in disordered polypeptides[28]. Peptides where opposing charge is equally mixed behave like random coils, while segregation of opposing charges leads to the formation of hairpin structures driven by long-range electrostatic interactions[28]. The biological implications of net charge in disordered regions can be exemplified by the complex formed between the human protein histone H1 and prothymosin-α. These disordered nuclear proteins have opposing net charges and bind with picomolar affinity, while retaining a significantly disordered state even in complex[29].

The compaction of the N-terminus of RNF4 positions the RING domain proximal to the SIMs to facilitate ubiquitin transfer (Fig. 6). This was highlighted through mutation of the arginine-rich motif leading to a more extended RNF4 shape and a delay in ubiquitination activity (Fig. 7). Intriguingly this delay in activity with the ΔBasic mutant seems to mostly affect the initial ubiquitin modifications, while the ubiquitin chain is short. Once growing ubiquitin chains have developed, RNF4 ΔBasic eventually functions at a rate similar to its WT counterpart (Fig. 7c). This is likely the result of longer ubiquitin chains being able to bridge the gap between SIMs and RING domain better. These observations probably explain why the phenotype of the ΔBasic RNF4 in vivo is intermediary between the wild-type protein and a null. Our observations highlight how proximity of the RING domains to the target lysine impacts the rate of ubiquitin transfer and also offers an explanation for how phosphorylation of RNF4 could control or modulate the rate of ubiquitination[30]. The activity of RNF4 promotes DNA double strand break repair via the homologous recombination pathway, with SUMOylation followed by RNF4-mediated ubiquitination of mediator of DNA damage checkpoint 1 (MDC1) required[3,4]. Intriguingly, phosphorylation of RNF4 at residues T26 and T112 by the cyclin-dependent kinase CDK2 in S-phase has been demonstrated to enhance its ubiquitination of MDC1[30]. In addition, a quantitative mapping of protein phosphorylation sites in rat organs identified S98/S99 as a phospho-site[31]. The S98/S99 and T112 phosphorylation would be positioned just after the arginine-rich motif in

RNF4 in the SIMs/RING-domain linker. As arginine-rich regions have been demonstrated to bind phosphate groups with covalent-like stability that can withstand mass spectrometric fragmentation[32], the phosphorylation of RNF4 would increase its acidic properties and enhance interactions with the RNF4 arginine-rich motif. This would stabilise compaction with RING and SIMs held close to rapidly discharge the ubiquitin cargo from the RING-bound E2~Ub to the SUMO chain.

Processivity is a phenomenon common to enzymes that act on polymeric substrates, such as proteins[33]. Ubiquitination is processive with multiple ubiquitin conjugation events taking place within a single E3/substrate encounter, demonstrated for both SCF$^{Cdc34}$ and SCF$^{β-TRCP}$ cullin RING ligases[34]. While the RNF4 RING domain is positioned close to the SIMs and this promotes substrate ubiquitination, the N-terminus remains flexible. This would explain how RNF4 can grow ubiquitin chains at multiple sites within SUMO and a variety of proteins to which SUMO is conjugated (such as PML)[6]. Additionally SUMOylation is often described as a spray, targeting multiple sites within a protein and multiple proteins within a complex[35]. In this context, the N-terminal region in RNF4 would allow efficient initiation of ubiquitin chain formation due to enhanced activity arising from its compaction, but the inherent flexibility would also allow RNF4 to processively grow long ubiquitin chains on both SUMO chains to which it is bound, and proteins across a more expansive area. Together this would facilitate a rapid amplification of the SUMO/Ub signal cascade (Fig. 8). As many ubiquitin E3 ligases or their targets contain regions shown or predicted to be disordered[9], this work provides a framework for understanding how highly charged regions increase the probability that the flexible substrate bound to the intrinsically disordered substrate adapter makes a productive encounter with the highly reactive ubiquitin-loaded E2 that is primed for catalysis on the RING. More generally it shows how charge segmentation allows dynamic and intrinsically disordered proteins to adopt ensembles containing preferred structures that are essential for function.

## Methods
**Cloning and production of recombinant proteins**. All RNF4 and RNF4N variants used were derived from *Rattus norvegicus* (98% sequence homology with human) containing C55S and C95S mutations. Full-length RNF4 variants (genes synthesized by GenScript) were expressed from the pLOU3 vector in Escherichia coli BL21 (DE3) cells, while RNFN 27-118 variants (gene fragments produced as gBlocks by IDT and recombined into pHis-TEV-30a using New England BioLabs HiFi assembly) were expressed from pHis-TEV-30a vector in E. coli Rosetta (DE3) cells as previously described[6,36]. $^{15}$N-enriched RNF4N 32–133 peptides (gBlocks recombined into pHis-TEV-30a using New England BioLabs HiFi assembly) were expressed from pHis-TEV-30a vector, while $^{15}$N-enriched RNF4 32–194 was expressed from the pLOU3 vector, all in E. coli Rosetta (DE3) cells as previously described[37]. For $^{15}$N-enriched peptides/proteins cells were grown at 37 °C in M9 minimal medium supplemented with $^{15}$NH$_4$Cl (Sigma). Samples were purified by Ni-NTA (Qiagen) chromatography following cleavage with TEV protease. RNF4N peptides were further purified to homogeneity by gel filtration (following biotinylation for single-molecule peptides).

Linear SUMO2 (genes synthesized by GenScript) genes bearing C47A mutations (4×SUMO) was subcloned into pHis-TEV-30a vector, along with a second variant containing a single cysteine at the C-terminus. Fusion proteins were purified as described for RNF4N peptides.

**Labelling proteins/peptides with functional groups**. RNF4N peptides for single-molecule measurements were biotinylated within a C-terminal AviTag using the E. coli biotin ligase BirA, while full-length RNF4 proteins were biotinylated at an N-terminal Avitag[38]. Biotinylation reactions contained 200 μM AviTag-fused RNF4N peptides/RNF4 proteins, 5 mM MgCl$_2$, 200 mM KCl, 2.5 mM ATP, 800 μM D-biotin in 50 mM Tris, 150 mM NaCl buffer at pH 7.5 (total volume 500 μl). Reactions were incubated at room temperature for 4 h, followed by overnight incubation at 4 °C. Biotinylated RNF4N/RNF4 was then purified by gel filtration and correct mass confirmed by LCMS (see examples in supplementary data). Next, RNF4N peptides were labelled stochastically with FRET dyes at a 2:2:1 molar ratio (Cy3B: Alexa647: RNF4N), in 50 mM Tris, 150 mM NaCl at pH 7 (or 1:1:1 molar ratio for dye labelling of full-length RNF4). FRET dyes used contained maleimide function groups for cysteine labelling, with dyes dissolved in DMSO (Cy3B, GE Healthcare; Alexa647,

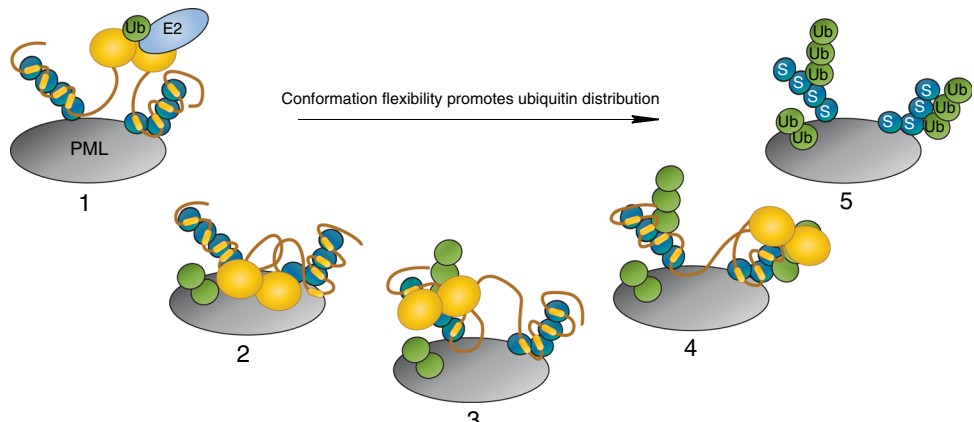

**Fig. 8 Compact but flexible RNF4N-terminus promotes ubiquitin distribution.** Step 1, PML protein (grey) is modified with SUMO (blue), which recruits RNF4 (orange). RNF4 dimerises through its C-terminal RING domains, activating it, before binding an E2~Ub conjugate (light blue, green respectively). Steps 2–4, the compact but flexible N-terminus of RNF4 allows the RING-dimer to access the area around the SIM bound SUMOs. Step 5, this broad distribution of ubiquitin promotes rapid amplification of the SUMO/Ub signal.

Invitrogen). Reactions were incubated at room temperature for 2 h before filtration to remove unreacted dyes using Centripure P2 filtration columns, into 50 mM Tris, 150 mM NaCl at pH 7.5 (or 45 min at 4 °C for dye labelling of full-length RNF4). Correct mass of labelled peptides was confirmed by LCMS (see examples in supplementary data). 4×SUMO fusion protein containing a single cysteine was labelled with Alexa488 (Invitrogen) as described for RNF4N/RNF4.

[15]N-enriched RNF4N peptides containing a single cysteine for MTSL labelling were stored under reducing conditions in 50 mM Tris, 150 mM NaCl, 0.5 mM TCEP at pH 7.5. Immediately prior to labelling reaction, peptides were buffer exchanged using a Centripure P2 filtration column into 50 mM Tris, 150 mM NaCl at pH 7.4. Reactions were performed at a 5:1 molar ratio of MTSL (dissolved in acetonitrile) over RNF4N for 1 h at room temperature. Unreacted MTSL was removed using a Centripure P2 filtration column into 50 mM Tris, 150 mM NaCl at pH 7.4, with correct mass of labelled peptide confirmed by LCMS.

**Native gel binding assay.** Due to the stochastic labelling of RNF4N peptides with FRET dyes, final concentration were only approximations. To assess binding to 4×SUMO native gel binding assays were performed[39]. RNF4N was mixed with 4xSUMO, final concentrations ~2 and 10 μM, respectively, and incubated at room temperature for 10 min in a total volume of 20 μl. Protein binding reactions were performed in 50 mM Tris, 150 mM NaCl, 0.5 mM TCEP at pH 7.5. Next, native loading buffer was added to samples resulting in a 10% (v/v) glycerol concentration, before loading onto 6% DNA retardation gels (Invitrogen). Samples were separated 60 V, 4 °C in 0.5× Tris-borate-EDTA buffer before analysis by in-gel fluorescence using a Bio-Rad ChemiDoc Imaging System.

**Molecular modelling of dye positions.** For the three-dimensional FRET models produced, the NMR structure PDB: 2MP2[12] was used as template for the kinked model, while the NMR structure PDB: 5M1U[40] was used as template for the extended model. The modelling of dyes was performed using FRET positioning software developed by the Seidel lab[19]. The software produces a sphere that approximates each dye defined by three radii, attached to the peptide by a flexible linker defined by its length (L-linker) and width (W-linker). For Cy3B: L-linker 20.5 Å, W-linker 4.5 Å, Rdye1 6.8 Å, Rdye2 3.0 Å, Rdye3 1.5 Å. For Alexa647: L-linker 21 Å, W-linker 4.5 Å, Rdye1 11 Å, Rdye2 4.7 Å, Rdye3 1.5 Å. The parameters are used by the software to calculate the dye accessible volume (AV) that simulates all accessible positions through manipulation of the linker from the point of attachment to the peptide. The attachment positions used for the kinked structure on PDB: 2MP2 are four residues shorter than those used in RNF4N 44/70 (23 and 27, respectively), due to limited access to residue sides chains with the structure. The attachment positions used for the extended structure PDB: 5M1U have the same residue separation as RNF4N 44/70. The modelled dyes were then used to produce predicted FRET efficiency values. This prediction uses the R0 of 60 Å previously reported for this FRET pair[18].

**Single-molecule total-internal reflection.** Single-molecule FRET (smFRET) experiments[41] were carried out as follows. smFRET intensity traces were acquired from immobilized peptides/proteins using a prism-type total-internal reflection setup that includes an inverted microscope (Olympus IX71) coupled to a 532-nm laser (Crystalaser) and a back illuminated Ixon EMCCD camera (Andor). Donor and acceptor fluorescence intensities were separated using dichroic mirrors (DCRLP645, Chroma Technology) and imaged onto the left (donor) and right (acceptor) half-chip of the EMCCD. This setup allowed us to monitor the Cy3B

and the Alexa647 signals simultaneously. Images were processed using IDL and data analysis was performed using laboratory-written routines in Matlab as previously described[42]. smFRET trajectories were acquired at 200 ms integration time unless stated otherwise. The FRET efficiency is then calculated as:$E = \frac{I_A}{\gamma I_D + I_A}$,, where $I_D$ represents the intensity of the donor (Cy3B) when exciting the donor and $I_A$ is the intensity of the acceptor (Alexa647) upon donor excitation. These signals are corrected for the leakage of the donor into the acceptor channel when exciting the donor, which is ~12% in our setup. The factor γ accounts for the differences in quantum yields and detection efficiency between the donor and acceptor channels, which is estimated as 0.92 in our setup[42]. When constructing FRET population histograms, the first ten frames of each movie were averaged to produce an average FRET value for that movie. Data were acquired in imaging buffer (50 mM Tris-HCl (pH 7.5), 150 mM NaCl, 6% (w/w) glucose, 0.1 mg/ml glucose oxidase (Sigma) and 0.02 mg/ml glucose catalase (Sigma), 1 mM Trolox. Cross-correlation analysis of single-molecule trajectories of donor and acceptor dyes within the same molecule was carried out using a home-made routine implemented in Matlab using the expression: $G(\tau) = \frac{\sum (I_D(t) - \overline{I_D})(I_A(t+\tau) - \overline{I_A})}{N \sum \overline{I_D I_A}}$ where $I_D(t)$ and $I_A(t)$ are the donor and acceptor intensities at a given time point. $\overline{I_D}$ and $\overline{I_A}$ represent the mean donor and acceptor intensities over the entire single-molecule trajectory with the function normalized to the total number of data points (N). The G(τ) function compares the intensity trace of the donor at a time t and the acceptor trace at a time t + τ.

**Paramagnetic relaxation enhancement (PRE) NMR spectroscopy.** RNF4N/RNF4 were dissolved in buffer containing 50 mM Tris-HCl, pH 7.4, 100 mM NaCl and 10% DMSO; RNF4 samples also contained 2 mM TCEP. A final concentration of ~150 μM [15]N-enriched RNF4N fragments were used for NMR measurements, and SUMO chains were added for a monomer ratio of 1:2 for RNF4N:SUMO in the complex studies. Standard [1]H–[15]N HSQC spectra were recorded for these MTSL-labelled samples, which were then reduced by adding 2 mM sodium ascorbate and incubated at 4 °C overnight, and [1]H–[15]N HSQC were recorded using the same settings on reduced samples. The intensity ratio were then converted into distance restraints[43]. A final concentration of ~300 μM [15]N-enriched RNF4 (32–194) was used, with TROSY-HSQC measurements performed. Experiments were run on a Bruker Avance HD III 800 MHz NMR spectrometer at 25 °C.

**Hydropathy plot of charge distribution.** Hydropathy plot was generated using online tool from Innovagen AB (Sweden), https://pepcalc.com/.

**Molecular dynamics simulations.** Initial fully extended conformations of RNF4N wild-type and basic mutant were generated using Modeller v9[44] and processed with the PDB2PQR server http://server.poissonboltzmann.org/pdb2pqr/[45]. These were then mapped to course-grained (CG) using SIRAH 2.0[46] The CG models were solvated using pre-equilibrated WT4 molecules in octahedral boxes of 2.0 nm size from the solute. An ionic strength of 0.15 M was set by randomly replacing WT4 molecules by Na$^+$ and Cl$^-$ CG ions. The system was prepared following standard SIRAH protocols: 5000 steps of steepest descent energy minimisation with positional restraints of 1000 kJ mol$^{-1}$ nm$^{-2}$ on backbone particles, followed by 5000 steps with no backbone restraints, followed by 5 ns NVT simulation at 300 K with 1000 kJ mol$^{-1}$ nm$^{-2}$ restraints on the protein to equilibrate the solvent, and a 25 ns simulation with reduced positional restraints of 100 kJ mol$^{-1}$ nm$^{-2}$ on backbone

beads to improve side chain solvation. Production simulations used the NPT ensemble at 300 K and 1 bar. Simulations were performed using GROMACS v5 (http://www.gromacs.org)[47]. PME electrostatics were used with the Verlet scheme with a cut off of 1.2 nm. Solvent and solute were coupled separately to V-rescale thermostats[48] with coupling times of 2 ps. Pressure was controlled by the Parrinello–Rahman barostat[49] with a coupling time of 8 ps. Production trajectories were performed for 1 μs. Trajectory analysis and visualisation used VMD[50] and gromacs tools. Residue-residue contact plots were generated using the mdmat tool for Cα atoms.

**Ubiquitination assays.** Ubiquitination assays were performed as described previously[51]. To measure total ubiquitination assays contained fluorescently labelled ubiquitin (Fluorescein, Invitrogen). Reactions were carried out by mixing the following components: 0.2 μM E1, 1 μM UbcH5a, 0.4 μM RNF4, 9 μM ubiquitin, 1 μM ubiquitin-Fluorescein, 3 mM ATP, 5 mM MgCl$_2$, 50 mM Tris, 150 mM NaCl, 0.5 mM TCEP. 4xSUMO substrate titration range: 0.125–8 μM. Reactions were quenched with SDS-PAGE loading buffer. Samples were subject to SDS-PAGE electrophoresis before analysis for in-gel fluorescence using a Bio-Rad ChemiDoc Imaging System. T = 0 samples were taken before the addition of ATP.

Direct 4xSUMO substrate ubiquitination assays were performed as described above with the following modifications. Ubiquitin was unlabelled and added at 20 μM. The 4×SUMO substrate was added at 1 μM, with 4×SUMO-Alexa488 at 0.25 μM.

Real-time ubiquitination assay were performed as described previously[52] on a PHERAstar (BMG labtech) via an FP optics module (FP 485 520). Assays were performed in a Greiner microplate (384 well, black, non-binding) and measured over the course of 50 min in a total volume of 20 μl. Reactions were carried out by mixing the following components: 0.1 μM E1, 1 μM UbcH5a, 0.4 μM RNF4, 9.6 μM ubiquitin, 0.4 μM ubiquitin-Fluorescein, 3 μM 4×SUMO, 3 mM ATP, 5 mM MgCl$_2$, 50 mM Tris, 150 mM NaCl, 0.5 mM TCEP.

**Lysine discharge assay.** Lysine discharge assays were performed as previously described[51]. E2 (Ubch5a) was first loaded with ubiquitin in the absence of a substrate or E3. The E2–Ub thioester was prepared by incubating 100 μM E2 with 114 μM ubiquitin and 6 μM ubiquitin-Fluorescein, along with 0.2 μM E1 in 50 mM Tris, 150 mM NaCl, 3 mM ATP, 5 mM MgCl$_2$, 0.1% NP40 at pH 7.5. The reaction was incubated at 37 °C for 12 min. To stop the loading of E2 with ubiquitin by E1, ATP was depleted using Apyrase (4.5 U/ml; New England BioLabs) at room temperature for 10 min. The E2–Ub thioester was then mixed at a 1:1 ratio with RNF4 and L-lysine in 50 mM Tris, 150 mM NaCl at pH 7.5 (final concentrations at 0.3 μM RNF4 and 25 mM L-lysine). The Reactions were incubated at room temperature before quenching using non-reducing SDS-PAGE loading buffer. Samples were subject to SDS-PAGE electrophoresis before analysis for in-gel fluorescence using a Bio-Rad ChemiDoc Imaging System. T = 0 was taken before the E2–Ub thioester was mixed with RNF4 and L-lysine

**Affinity assay by fluorescence polarization.** The binding affinity between RNF4 and 4×SUMO was performed via fluorescence polarization with settings as described for real-time ubiquitination assay. Unlabelled RNF4 was titrated against 1 μM 4×SUMO-Alexa488 in 50 mM Tris, 150 mM NaCl, 0.5 mM TCEP at pH 7.5. Reactions were incubated at room temperature for 10 min before measuring FP.

**Cell methodology.** U2OS RNF4$^{-/-}$ cells, generated in-house[53], were transfected with a flag-tagged version of RNF4 WT or RNF4 ΔBasic using lipofectamine 3000 and stable cells selected with 6 μg/ml puromycin. Western blotting for RNF4 used an in-house antibody at 1:10,000 dilution, raised in chicken as described[53]. Immunofluorescence analysis was carried out on cells fixed with 4% formaldehyde and permeabilized with 0.1% triton X-100 as described[54] using a monoclonal antibody to RNF4[55] used at 1:1000 dilution and an in-house antibody to PML raised in chicken[54] used at 1:5000 dilution. Arsenic trioxide in medium was used at a final concentration of 1 μM for the indicated times.

**Reporting summary.** Further information on research design is available in the Nature Research Reporting Summary linked to this article.

## Data availability

Source data are provided with this paper and can be accessed at https://doi.org/10.17630/5d766785-60c8-4dbe-99ec-926e2fa33798. All other data are available from the corresponding author on reasonable request. Source data are provided with this paper.

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

## Acknowledgements

This work was supported by the following grants: Wellcome Trust Investigator Awards (098391/Z/12/Z and 217196/Z/19/Z) and Cancer Research UK Programme grant (C434/A21747) to R.T.H., Wellcome Trust Studentship (109113/Z/15/Z) to P.M., Wellcome Trust Collaborative Award (215539) and multiuser equipment grant (104833) to S.J.M. Additionally J.C.P. thanks the Scottish Universities Physics Alliance (SUPA) and the University of St. Andrews for financial support. We would also like to thank Michael Tatham for his contribution to data analysis and Federico Pelisch for his feedback on the manuscript. In addition, we would like to acknowledge Professor Takeshi Urano (Shimane University School of Medicine, Japan) for kindly providing the monoclonal antibody to RNF4.

## Author contributions

P.M. cloned expressed and purified proteins, conducted smFRET, smFRET modelling, biochemical experiments and performed data analysis. J.C.P. conducted smFRET experiments and performed data analysis. S.J.M. designed NMR experiments and contributed to data analysis. Y.X. conducted NMR experiments and performed data analysis. A.P. cloned expressed and purified proteins, conducted NMR experiments and performed data analysis. S.L.R. conducted molecular simulations and performed data analysis. E.G.J conducted in vivo experiments and performed data analysis. R.T.H conceived the project and contributed to data analysis. All authors contributed to manuscript preparation.

## Competing interests

The authors declare no competing interests.
