## [Peer Review File · Nature Communications]

Reviewers' comments:

Reviewer #1 (Remarks to the Author):

The manuscript by Murphy et al (Ron Hay and coworkers) reports an important discovery regarding the architecture of a RING E3 ligase that suggests how substrate recruitment domains can be spatially positioned in close proximity to the RING domain. Importantly this paper suggests that intrinsically disordered regions can have a key role in domain organisation.

The manuscript is well written, concise and highlights the key findings. It also makes a key contribution to the field as it is likely that other E3s might have similar properties – this is something that it will be important to test.

As a result publication is supported.

Main point:

1) Previously the authors have shown that dimerisation of the RNF4 RING domain is important for activity. Have the authors evaluated the impact of RING dimerisation on the compaction of RNF4? It would be nice to see how an obligate RNF4 monomer behaves in these assays?

It seems that a constitutive monomeric form of RNF4 would also be compact prior to addition of SUMO chains but this will be important to understand as it will inform the order of events associated with activation of RNF4.

2) Furthermore, is the SUMO chain ubiquitylated in an intramolecular manner? It seems that this model would predict that to be the case but it would be nice if the authors could demonstrate this.

Minor point:

It might be helpful if the authors included a schematic to sum up their data.

Reviewer #2 (Remarks to the Author):

The authors employ single-molecule FRET and NMR to study the conformation and flexibility of a protein linker of the E3 ligase RNF4N in the context of ubiquitin transfer. For a substrate recognition domain previously thought to be intrinsically disordered, the authors report structurally defined conformations. This defined structural architecture might indicate a general mechanism of this family of E3 ligases.

First, I have a couple of questions regarding fluorophore labeling of the RNF4N peptides. According to the methods section, peptides were purified using filtration columns, which will remove the excess dye. However, this purification will also retain the unlabeled peptide. Did the authors determine the degree of labeling for the peptides? And, do I correctly interpret that “stochastic labeling” (Methods) refers to the random labeling of dual-cysteine mutants of the RNF4N peptides with either fluorophore? I would expect a significant fraction of peptides labeled with two donors and two acceptors (in theory, each would be 25%); did the authors analyze the distribution of dual-labeled peptides?

Second, I ask the authors to include more details on the FRET analysis. How were single-molecule spots pre-selected? The authors averaged intensities from the first ten frames; an alternative (or additional) analysis would plot the FRET values for each 100 ms interval, and for complete trajectories until acceptor/donor photobleaching.

Third, the authors show smFRET trajectories of the WT peptide (Fig 1E and SI figures). There seems to be no apparent dynamics (in a sense of conformational transitions resulting in distance changes); however, the width of the FRET populations (Fig 1C) seems to be different, which could report on some intrinsic flexibility in part of the regions. This information could be extracted in addition, and would allow to refine the structural model.

Minor issues

Please use the term “Förster resonance energy transfer”, according to IUPAC recommendation, <https://doi.org/10.1351/pac200779030293>

Figure 1C, do the numbers in brackets indicate the position of the fluorophore labels?

Single-molecule FRET experiments operate at low concentrations compared to biomolecular interaction strengths. What concentrations were used in the single-molecule assay? Can the authors comment on the K_D of SUMO binding the peptide? One could also think of ensemble FRET experiments at concentrations as high as in the shift assays.

Reviewer #3 (Remarks to the Author):

This manuscript from Murphy et al. aims to understand how SUMO-chains are being ubiquitinated by the E3 ligase RNF4. This follows their long-standing biological (Tatham et al. (2008); Yin et al. (2012)) and mechanistic (Plechanovova et al. (2011), Plechanovova et al. (2012); Xu et al. (2014)) interest in the actions of the SUMO-targeted RNF4.

In this particular work, the authors use FRET to show that SUMO binding does not induce substantial conformational change in the SIM domains of RNF4 upon binding (Figure 1). The authors also show by FRET that mutation of basic residues between the SIM domains and the RING causes the protein to be less compact (Figure 2). NMR and MD simulations further support this FRET data (Figure 3). Finally, mutation of the basic residues reduces in vitro ubiquitination of a SUMO substrate (Figure 4). The biophysical data is of good quality and the fact that the SIMs are ready to engage substrate is of interest.

However, although the authors attempt to tackle one of the big questions in the ubiquitin field, there are many issues with the presented work. As a study, it feels like one-half of a story and lacks detailed mechanistic insight and physiological relevance. On the mechanistic side, there is little explanation or data to explain why the basic residues are important for function: What are they doing to facilitate compaction of the SIMs? What are they interacting with? The Δ basic mutant appears to be important for compaction of the ligase, which brings substrate and RING together. However, there is no mechanism presented here that explains why and how. There is also no data that physically show that the RING domains are actually close to the substrate binding SIMs. There seem to be acidic patches in the sequence. Are they interacting with the basic region? Without addressing these questions all we know is that mutating RNF4 perturbs its structure and function.

On the physiological side, some cellular data would help demonstrate that the decrease in activity resulting from mutation of the basic residues is functionally relevant. The authors have previously characterized and measured the physiological functions of RNF4 in cells (Tatham et al. (2008); Yin et al. (2012)). It is therefore surprising that they didn't test in this study whether their in vitro findings are important for RNF4's cellular function.

In addition, the following issues were found:

- SUMO Binding in Extended data Fig 12b looks impaired in the gel shift assay in contrast to what is stated in the main text (p4 l86 “retained SUMO chain binding”).
- The table in extended data table 4 suggests that the end-to-end distance is relatively similar between WT and Δ basic. (8.4 vs. 10.1). These values are both within the standard deviation of the measurements. Is there now a significant opening in the Δ basic mutant or not?
- The NMR experiments with N- or C-terminal spinlabels hardly show an effect. Why are only residues 74 and 75 slightly affected and what about the rest of the construct?
- The manuscript is very short and there is no discussion. How do the results change our view of RNF4 mechanism and function? How does this contrast with our understanding of other ligases? For instance, one interesting comparison might be Cullin-RING E3s and how they achieve substrate ubiquitination (Substrate adaptor on one end, RING on the other, activation by Nedd8 so that the RING can span the distance to the substrate (e.g. Duda et al. (2008) Cell).
- No data in the paper shows that the RINGs are brought in close proximity to the SIMs and thus the substrate.
- Generally this manuscript makes it hard to find information about constructs and more. The Figures are not intuitive and proper labelling would make it much easier to understand them (e.g. the little cartoons above the PRE NMR data have the wrong size and are therefore confusing. These cartoon in the correct size could show where features are in the histogram.)

Reviewer #4 (Remarks to the Author):

In this manuscript, Murphy et al. showed that a defined architecture within a disordered region contributes to E3 ligase function. The authors discovered that the expected disordered substrate recognition domain of RNF4 might have a defined conformation before SUMO binding using the smFRET and NMR methods. They further mutated the highly basic linker. Using smFRET and molecular dynamics simulations, the authors proposed that electrostatic interactions may contribute to the formation of the highly flexible but compact structure. This manuscript addressed a significant question – what is the mechanism by which SUMO substrate is delivered to the RING - by proposing an electrostatic driven compact structure within a disordered region. Overall, I am impressed by the quality of the paper. However, the proposed mechanism still suffers from multiple deficiencies. My major concerns are that the proposed electrostatic interactions and the global three-dimensional architecture are not well defined in the manuscript. Thus, I think the paper is a bit immature to be accepted as it is. If the authors could address the points discussed below, it could be a very significant and interesting paper.

Major concerns

1. One of the major discoveries of the paper is that “electrostatic interactions involving the highly basic region linking the substrate recognition and RING domains juxtaposed those regions and mediated substrate ubiquitination”. However, it is not clear what residues the basic region interacts with? Is there an acidic region? Or are there any specific acidic residues involved in the electrostatic interactions with the basic region? Are these residues located in any specific SIM region? These questions are important to define the proposed electrostatic interactions.

2. The smFRET focuses on the conformation of the SIM region and demonstrated that the conformation of the SIM region may be primed for SUMO binding. The NMR data, on the other hand, suggested that both termini fold back towards the middle of the peptide. From Figure 3D and Extended data Figure 19, it seems that not only the basic linker, other regions, including residue ~105-115 may also have interactions with other residues. But these interactions are not discussed in the manuscript. Could the interactions involving residues ~105-115 be important for the highly flexible but compact structure? Could the authors use smFRET to measure the conformational change between the basic linker and the RING domain (i.e. the C-terminal region, residues 87-133) and to show whether the conformation of this region is defined or disordered before and after SUMO binding? Or could the mutagenesis experiments demonstrate the roles of residues ~105-115?

3. The authors stated that their studies “have revealed a global three-dimensional architecture within this “disordered” region, with highly flexible but compact structures facilitating productive binding of SUMO substrate.” However, it is not clear how this 3D architecture is defined. Does the entire peptide (residue 32-133) or only SIM+basic linker region (residue 32-88) have a 3D architecture? What characterizes the compact structures?

4. The molecular dynamics simulations showed the distances between residues 80 and 133 change upon the mutation. However, in Extended data Table 4, the authors did not report distances change between residues 32-79 upon mutation. Are these distances change significantly upon mutation? If not, what could be the reason if the basic residues are critical to maintaining the compact structure in the N-terminal region?

Minor Concerns

1. The basic residues are mutated to either serine or alanine. Why are they mutated to different residues?

2. On Page 18, "Production trajectories were performed for." This sentence is not completed. How long did the production trajectories were performed for?

We would like to thank the reviewers for their constructive criticism of our paper. The additional experiments, suggested by the reviewers, that we have carried out and incorporated into the revised manuscript have certainly improved our paper.

Reviewers' comments:

Reviewer #1 (Remarks to the Author):

The manuscript by Murphy et al (Ron Hay and coworkers) reports an important discovery regarding the architecture of a RING E3 ligase that suggests how substrate recruitment domains can be spatially positioned in close proximity to the RING domain. Importantly this paper suggests that intrinsically disordered regions can have a key role in domain organisation.

The manuscript is well written, concise and highlights the key findings. It also makes a key contribution to the field as it is likely that other E3s might have similar properties – this is something that it will be important to test.

As a result publication is supported.

Main point:

1) Previously the authors have shown that dimerisation of the RNF4 RING domain is important for activity. Have the authors evaluated the impact of RING dimerisation on the compaction of RNF4? It would be nice to see how an obligate RNF4 monomer behaves in these assays?

It seems that a constitutive monomeric form of RNF4 would also be compact prior to addition of SUMO chains but this will be important to understand as it will inform the order of events associated with activation of RNF4.

This is indeed an important point to address. Thus we include an NMR experiment (Fig. 1 revised manuscript) where we compare the monomeric N-terminal region of RNF4 (RNF4N; 32-133) with RNF4 (32-194), that contains the RING domain. Under the conditions of the NMR experiment (200-300 μ M concentration) the RNF4 will be dimeric as this is more than 1000 times higher than the k_D for dimerization (180 nM). In this experiment identical shifts are apparent for residues in the N-terminal region from both monomeric RNF4N (32-133) and dimeric RNF4 (32-194). This indicates that the compaction observed in the RNF4N ensemble is also present in full-length RNF4 (see text p4, line 13-23)

2) Furthermore, is the SUMO chain ubiquitylated in an intramolecular manner? It seems that this model would predict that to be the case but it would be nice if the authors could demonstrate this.

We did not address this point in this manuscript, but have done so in a previous publication (Plechanovova, A., Jaffray, E. G., McMahon, S. A., Johnson, K. A., Navratilova, I., Naismith, J. H., and Hay, R. T. (2011) Mechanism of ubiquitylation by dimeric RING ligase RNF4. *Nature structural & molecular biology* **18**, 1052-1059). We demonstrated that RNF4 ubiquitination of bound SUMO can proceed in both *cis* and *trans* from either RNF4 molecule.

Minor point:

It might be helpful if the authors included a schematic to sum up their data.

In response to this we have added a model with each data set to highlight what each figure represents, with a final combined model at the end.

Reviewer #2 (Remarks to the Author):

The authors employ single-molecule FRET and NMR to study the conformation and flexibility of a protein linker of the E3 ligase RNF4N in the context of ubiquitin transfer. For a substrate recognition domain previously thought to be intrinsically disordered, the authors report structurally defined conformations. This defined structural architecture might indicate a general mechanism of this family of E3 ligases.

1) First, I have a couple of questions regarding fluorophore labeling of the RNF4N peptides. According to the methods section, peptides were purified using filtration columns, which will remove the excess dye. However, this purification will also retain the unlabeled peptide. Did the authors determine the degree of labeling for the peptides?

Following dye labelling of the RNF4N peptides LCMS analysis was performed. In all peptides labelled no parent (unlabelled) material was detected (Supplementary Fig 1). In the newly added experiment where full length RNF4 was labelled within the RING domain, the dye/protein ratio, and time of reaction were both reduced to ensure zinc coordinating cysteines were not modified. As a result there was some under-labelling in this case (donor/acceptor only) as detailed in the supplementary data (Supplementary Fig 17). However this not an issue as only molecules containing 2 FRET dyes were selected during analysis (text p16, line 24-p17, line 8)

2) And, do I correctly interpret that “stochastic labeling” (Methods) refers to the random labeling of dual-cysteine mutants of the RNF4N peptides with either fluorophore? I would expect a significant fraction of peptides labeled with two donors and two acceptors (in theory, each would be 25%); did the authors analyze the distribution of dual-labeled peptides?

The reviewer is correct, stochastic labelling refers to the random incorporation of fluorophores at the two cysteine mutants in each peptide. If the reactivity of both dyes was the same for each of the two cysteines, their relative populations will be 25% as indicated by the reviewer. However, we observed a slightly bias of the labelling efficiency towards the Cy3B dye during stochastic labelling (text in legend to Supplementary Fig 17). In addition to mass spec, we also quantified the ratio of immobilized molecules showing D-A pairs and D-D pairs when exciting the donor. As an example, 600 molecules analysed from the SIM1-2 construct showed a (D-D):(D-A) ratio of 2:1, suggesting a more efficient reaction for the Cy3B dye. However, given that we are working at the single-molecule level and we can select those constructs carrying a single D-A pair, and as such differences in labelling ratios have no impact on the data. For example see supplementary Fig. 1D).

3) I ask the authors to include more details on the FRET analysis. How were single-molecule spots pre-selected?

Spot selection was carried out using a custom-written software in IDL (Research Systems, Boulder, CO) that was initially developed in the lab of Prof Taekjip Ha (University of Illinois-Urbana) as described elsewhere (R. Roy, S. Hohng, S., T. Ha, 2008 Nat. Methods, 5, 507-516). Briefly, individual spots were selected, and D-A pairs correlated as follows: First, a calibration is made using 200 nm fluorescent beads with signal in the donor and acceptor spectral range to provide a mapping algorithm that correlates spots on the left channel with acceptor spots on the right channel of the

EMCCD. Then, donor and acceptor images are superimposed to generate a total intensity image. After background image correction, total intensity spots are selected based on three criteria: i) the size is only 3-4 pixels, ii) the point-spread function (PSF) of each spot is fitted to a 2D gaussian shape and iii) the intensity of the spot is higher than a threshold value. With these combined intensity and PSF-shape criteria, we ensure that the spot correspond to single molecules (at least within the diffraction limit resolution) and not clusters and this is further filtered by observing single photobleaching steps and anticorrelated fluctuation of D and A signals. Once total intensity spots have been identified, the software uses the mapping algorithm to spatially correlate D and A signals coming from the same single-molecule and generates the intensity vs time trajectories for each molecule. The FRET efficiency is then calculated as:

$$E = \frac{I_A}{\gamma I_D + I_A}$$

where I_D represents the intensity of the donor (Cy3) when exciting the donor and I_A is the intensity of the acceptor (Alexa647) upon donor excitation. These signals are corrected for the leakage of the donor into the acceptor channel when exciting the donor which is ~10% in our setup. The factor γ accounts for the differences in quantum yields and detection efficiency between the donor and acceptor channels which is estimated as 0.92 in our setup (text p19, line 1-19).

4) The authors averaged intensities from the first ten frames; an alternative (or additional) analysis would plot the FRET values for each 100 ms interval, and for complete trajectories until acceptor/donor photobleaching.

We agree with the reviewer that there are several alternatives to describe the equilibrium populations of FRET states from single-molecule data. In this paper, we extracted an average FRET value per molecule using the first 10 frames of each single-molecule trajectory. This approach has been widely used by us (Heppell et al, 2011, Nat. Chem. Biol, 7, 384; McCluskey et al, 2019, Nucleic Acids Res. 47, 6478-6487) and others (Lee SJ and Ha. T, Methods. Mol. Biol, 2018, 1805: 233-250; Uhm, H et al. 2018, 115, 331-336) when using smFRET to investigate conformational changes in biomolecules. To demonstrate that this approach and the alternatives suggested by the reviewer provide similar quantitative results, we compared, as an example, the smFRET histogram of FRET populations obtained for the RNF4N (30-57) construct (Figure 2C) using the 'first 10 frames' method and the entire trajectory until donor/acceptor photobleaching (light blue) method indicated by the reviewer. The results are shown in Figure A (below). The FRET distribution of populations is very similar in both cases and this is further confirmed when quantitatively comparing the results of gaussian fitting to both histograms (see table below).

Figure A. Comparison of single-molecule FRET population histograms obtained for RNF4N (30, 57) using the ‘first 10 frames criteria’ (black) (Figure 2C) and using entire traces until photobleaching (light blue).

Analysis method	Low FRET gaussian centre	Low FRET gaussian width	High FRET gaussian centre	High FRET gaussian width
First 10 frames	0.298	0.07	0.661	0.14
Entire trace	0.305	0.07	0.656	0.15

5) Third, the authors show smFRET trajectories of the WT peptide (Fig 1E and SI figures). There seems to be no apparent dynamics (in a sense of conformational transitions resulting in distance changes); however, the width of the FRET populations (Fig 1C) seems to be different, which could report on some intrinsic flexibility in part of the regions. This information could be extracted in addition, and would allow to refine the structural model.

As noted by the reviewer and stated in the manuscript, most single molecule trajectories display a single FRET state and only a very small percentage (<1%) showed interconversion between high and low FRET conformations. We also agree with the reviewer that variations in the width of the FRET distribution might reflect either dynamic changes in distance or static heterogeneity within the sample. Thus, even a gaussian population assigned to a single FRET state might in fact represent an ensemble of structures dynamically exchanging at rates close to our time resolutions.

To investigate this further, we have calculated the cross-correlation function between the donor and acceptor signals of representative FRET trajectories displaying a ‘single’ FRET level. The shape of the cross-correlation function should reveal the presence of any dynamics hidden within the signal between structurally close conformers. Cross-correlation analysis of smFRET data has been previously used to extract the interconversion dynamics between structural states interconverting at rates close to the temporal resolution of the microscope (Wang J. et al, 2012, Biophys. J., 103, 2541-2548; Tan, E. et al 2003, PNAS 100, 9308-9313)

Here, we used two different set of data that are representative of narrow and wide FRET distributions. As examples of narrow distributions in both high and low FRET states, we used the construct RNF4N 30/57 whose histogram is shown in Figure 2B of the revised manuscript and in Figure B below (left panel). As example of a wide FRET distribution we used the construct RNF4N Δ Basic 57/84 (Supplementary Figure 12 in the revised manuscript and Figure B below, right panel). The results from the cross-correlation analysis of these data are shown in the figure below for both constructs:

Figure B. (Left) Single-molecule FRET histogram obtained for RNF4N 30, 57 (top panel), representative low- and high-FRET trajectories (middle panel) and cross-correlation curves (bottom panels) obtained for the traces shown in the middle panels. (Right) Single-molecule FRET histogram obtained for RNF4N Δ Basic 57,84 (top panel), representative low- and high-FRET trajectories (middle panel) and corresponding cross-correlation curves (bottom panel).

The $G(\tau)$ curves (bottom panels) exhibit a cross-correlation function that is horizontal and randomly distributed around zero for both constructs. This confirms the absence of dynamic correlation between the donor and acceptor emission down to the timescale of the data collection. Based on this, we concluded that single-level FRET states represent static conformations, at the least, within the time-resolution of the measurement (200 ms) and that FRET populations showing a wider distribution mostly reflect static heterogeneity within the sample between conformers with relatively similar, but not identical, structures.

Methods: Correlation-analysis of single-molecule trajectories of donor and acceptor dyes within the same molecule was carried out using a home-made routine implemented in Matlab using the expression:

$$G(\tau) = \frac{\sum (I_D(t) - \overline{I_D}) (I_A(t + \tau) - \overline{I_A})}{N \sum \overline{I_D} \overline{I_A}}$$

where $I_D(t)$ and $I_A(t)$ are the donor and acceptor intensities at time a given time point, $\overline{I_D}$ and $\overline{I_A}$ represent the mean donor and acceptor intensities over the entire single-molecule trajectory and the function has been normalized to the total number of data points (N). The $G(\tau)$ function compares the intensity trace of the donor at a time t and the acceptor trace at a time t+ τ (text p6, line 1-9).

Minor issues

1) Please use the term “Förster resonance energy transfer”, according to IUPAC recommendation, <https://doi.org/10.1351/pac200779030293>

We thank the reviewer for noticing this and we have modified the manuscript by replacing any ‘fluorescence resonance energy transfer’ term by ‘Förster resonance energy transfer’ (text p3, line 23-24)

2) Figure 1C, do the numbers in brackets indicate the position of the fluorophore labels?

Correct, the numbers in brackets represent the labelling position of the Cy3 (donor) and Alexa647 (acceptor) in this order. This has been clarified in the figure legends of the manuscript.

3) *Single-molecule FRET experiments operate at low concentrations compared to biomolecular interaction strengths. What concentrations were used in the single-molecule assay? Can the authors comment on the K_D of SUMO binding the peptide?*

The reviewer is correct that single-molecule fluorescence-based assays have a concentration limit whose value (~50-100 nM) depends on the specific nature of the label and microscopy detection method used. However, this limitation applies only when using fluorescently labelled molecules that are expected to bind the immobilized partner. In our case, we investigate the structural state of the substrate using an intra-molecular FRET approach, and therefore, there is no limitation in the amount of unlabelled SUMO we can add to the immobilized sample and we have used saturating concentrations of ~ 25 μ M (K_D for binding ~2 μ M) .

4) *One could also think of ensemble FRET experiments at concentrations as high as in the shift assays.*

We agree with the reviewer that ensemble experiments could be done with our FRET-labelled constructs and unlabelled SUMO; however, these measurements although possible will not be straightforward given the stochastic labelling of each construct. In contrast, the unique ability of single-molecule to select substrates that carry the appropriate D-A FRET pair simplifies the methodology and allow us to quantify the distribution of FRET populations for each construct that cannot be obtained from conventional ensemble FRET.

Reviewer #3 (Remarks to the Author):

This manuscript from Murphy et al. aims to understand how SUMO-chains are being ubiquitinated by the E3 ligase RNF4. This follows their long-standing biological (Tatham et al. (2008); Yin et al. (2012)) and mechanistic (Plechanovova et al. (2011), Plechanovova et al. (2012); Xu et al. (2014)) interest in the actions of the SUMO-targeted RNF4.

In this particular work, the authors use FRET to show that SUMO binding does not induce substantial conformational change in the SIM domains of RNF4 upon binding (Figure 1). The authors also show by FRET that mutation of basic residues between the SIM domains and the RING causes the protein to be less compact (Figure 2). NMR and MD simulations further support this FRET data (Figure 3). Finally, mutation of the basic residues reduces in vitro ubiquitination of a SUMO substrate (Figure 4). The biophysical data is of good quality and the fact that the SIMs are ready to engage substrate is of interest.

However, although the authors attempt to tackle one of the big questions in the ubiquitin field, there are many issues with the presented work. As a study, it feels like one-half of a story and lacks detailed mechanistic insight and physiological relevance.

1) On the mechanistic side, there is little explanation or data to explain why the basic residues are important for function: What they are doing to facilitate compaction of the SIMs? What are they interacting with? There seem to be acidic patches in the sequence. Are they interacting with the basic region?

To address this point we focussed on analysing the charge distribution of RNF4. A hydropathy plot (Fig. 4A in the revised manuscript) highlights the segregated charged regions within RNF4 (acidic SIM region, basic arginine-rich motif and acidic region in the SIMs/RING domain linker. Along with the Δ Basic mutant, we generated an RNF4N peptide variant with the acidic region removed from the SIMs/RING linker was produced (Δ Acidic). These peptides were assessed via size exclusion chromatography and smFRET. The loss of charge from either peptide caused altered elution profiles and altered FRET efficiency populations compared with the WT RNF4N peptide, highlighting the role for these charged regions in promoting compaction (text p8, line 5-p9, line 6). MD simulations were then employed to assess the potential for electrostatic interactions between the different charged regions. Results in this case indeed indicated that the basic arginine-rich motif engaged in intramolecular electrostatic interactions with its flanking acidic regions, with these interactions promoting compaction (text p9, line 7-23). In this respect in our revised discussion we refer to a report on segregated charge distribution in disordered regions contributing to hairpin structures (Das, R. K., and Pappu, R. V. 2013 Conformations of intrinsically disordered proteins are influenced by linear sequence distributions of oppositely charged residues. *Proceedings of the National Academy of Sciences of the United States of America* **110**, 13392-13397), similar to what we observe with RNF4 (text p13, line 23 – p14, line 2).

2) The Δ basic mutant appears to be important for compaction of the ligase, which brings substrate and RING together. However, there is no mechanism presented here that explains why and how. There is also no data that physically show that the RING domains are actually close to the substrate binding SIMs. Without addressing these questions all we know is that mutating RNF4 perturbs its structure and function.

The reviewer is correct in that we had no direct evidence of RING domain interacting with the SIMs as per our original model. To address this point we generated a full length version of RNF4 bearing a FRET pair positioned in the SIMs (residue 70) and RING domain (residue 144). A surface exposed serine residue in the RING domain was mutated to cysteine to allow conjugation of a maleimide functionalized dye. The integrity of the labelled RING domain was demonstrated through its ligase activity (Supplementary Fig. 17). smFRET analysis of this protein revealed direct interaction with the SIMs, high FRET (compact) and low FRET (extended) conformations. In addition and in line with our original model, loss of the basic charge from the arginine-rich motif from the SIMs/RING linker caused a significant reduction in the high FRET molecule population (text p10, line 1-21).

We would like to thank the reviewer for this suggestion as we feel this experiment adds significantly to the manuscript. We have now shown for a full-length and functionally active RNF4 direct interaction between SIMs and RING domain, and highlighted how the arginine-rich motif promotes interaction between these two key protein domains.

3) On the physiological side, some cellular data would help demonstrate that the decrease in activity resulting from mutation of the basic residues is functionally relevant. The authors have previously characterized and measured the physiological functions of RNF4 in cells (Tatham et al. (2008); Yin et al. (2012)). It is therefore surprising that they didn't test in this study whether their in vitro findings are important for RNF4's cellular function.

To address this point we used an RNF4 Δ - cell line (Maure, J. F., Moser, S. C., Jaffray, E. G., A, F. A., and Hay, R. T. (2016) Loss of ubiquitin E2 Ube2w rescues hypersensitivity of RNF4 mutant cells to DNA damage. *Scientific reports* **6**, 26178) and reintroduced either WT or Δ Basic versions of RNF4. Stable cell lines expressing equivalent levels of RNF4 WT and RNF4 Δ Basic were then analysed for the ability to induce degradation of the promyelocytic leukaemia protein (PML) in response to arsenic treatment (Tatham, M. H., Geoffroy, M. C., Shen, L., Plechanovova, A., Hattersley, N., Jaffray, E. G., Palvimo, J. J., and Hay, R. T. (2008) RNF4 is a poly-SUMO-specific E3 ubiquitin ligase required for arsenic-induced PML degradation. *Nature cell biology* **10**, 538-546). We assessed PML degradation by counting the number of PML bodies per cell in response to arsenic. As we have described previously PML body numbers increase after 1 hour of arsenic treatment and then decline over time. The data we collected indicated that when compared to WT RNF4 the degradation of PML was impaired in cells expressing the Δ Basic form of PML (Supplementary Fig. 18, text p12, line 11-23).

In addition, the following issues were found:

4) SUMO Binding in Extended data Fig 12b looks impaired in the gel shift assay in contrast to what is stated in the main text (p4 l86 "retained SUMO chain binding").

Due to the loss of the basic charge from the arginine-rich motif these peptides migrated differently than their WT counterparts during the native-PAGE electrophoresis. Since we have shown with the full length protein via fluorescence polarization that Δ Basic mutations do not impair binding (Fig. 6H), we have removed this figure from the supplementary data.

5) The table in extended data table 4 suggests that the end-to-end distance is relatively similar between WT and Δ basic. (8.4 vs. 10.1). These values are both within the standard deviation of the measurements. Is there now a significant opening in the Δ basic mutant or not?

The compactness (radius of gyration) and end-to-end distances increase for the Δ Basic mutant as expected, however, as noted by the reviewer, this type of analysis is limited due to the large variation in conformation across simulations for this type of protein. To include further insights into this behaviour we have now incorporated new analysis based on conformational clustering of the final 100 ns of each trajectory. This highlights the increased conformational flexibility of the extended conformations compared to the more compact configurations. We are also able to define 'loop' and 'hairpin' higher order structures based on the contact plots (Figure 4E-F, Supplementary figure 16, text p9, line 7-23).

6) The NMR experiments with N- or C-terminal spin-labels hardly show an effect. Why are only residues 74 and 75 slightly affected and what about the rest of the construct?

We have been very conservative in quoting values for RNF4N PREs i.e. amides have to be unambiguously assigned, possess good linewidths and be fully resolved so values can be calculated accurately. We quoted effects in the text for 74 and 75 as they lie significantly outside any experimental error. Furthermore, consistent with the notion of compacting on the molecular ensemble of RNF4N we observed exchanging broadening for residues within and around the basic region (between 78-81) therefore these peaks were either absent or could not be assigned, so PREs are not available for this region. This is unfortunate as it is likely that significant PREs would present. However, if we are less strict about our criteria we can comment on effects of other residues. For example, the PREs for residue 73 and 83 either side of the basic region are 0.85 and 0.77 respectively when the spin labels are present at the termini. Perhaps more notably, residue 77 is detectable in the control NMR spectrum but exchange broadened (less half the intensity of other amides) and this completely disappears in the paramagnetic state, suggesting a short distance.

With regard to the comment that these values represent only slightly affected residues, we believe they are consistent with the smFRET data. NMR PREs are sensitive to distances up to 30Å the spin label side and RNF4N amide, where a value of 1 represents a distance higher than this. Values of 0.7-0.8 correspond to distances less than 20Å, and residues 77 would be even shorter than this (text p7, line 1-20).

7) The manuscript is very short and there is no discussion. How do the results change our view of RNF4 mechanism and function? How does this contrast with our understanding of other ligases? For instance, one interesting comparison might be Cullin-RING E3s and how they achieve substrate ubiquitination (Substrate adaptor on one end, RING on the other, activation by Nedd8 so that the RING can span the distance to the substrate (e.g. Duda et al. (2008) Cell).

We agree with the reviewer and such have expanded on the introduction and discussion. We now refer to the recent mechanism revealed for Cullin-RING E3s whereby Neddylation of the cullin scaffold brings the RING domain and substrate into a proximity that promotes Ub transfer (Baek, K., Krist, D. T., Prabu, J. R., Hill, S., Klugel, M., Neumaier, L. M., von Gronau, S., Kleiger, G., and Schulman, B. A. (2020) NEDD8 nucleates a multivalent cullin-RING-UBE2D ubiquitin ligation assembly. *Nature* **578**, 461-466). It is worth noting however that these large multi-subunit E3s may not offer a direct comparison to an E3 like RNF4. As such, we focus on single-subunit E3s and reveal how prevalent regions of disorder are within this E3 subgroup. This highlights how our finding with RNF4 could offer insight into Ub transfer for hundreds of other single-subunit E3s, and how PTM such a phosphorylation could be used to regulate E3 activity (text p13-15).

8) Generally this manuscript makes it hard to find information about constructs and more. The Figures are not intuitive and proper labelling would make it much easier to understand them (e.g. the little cartoons above the PRE NMR data have the wrong size and are therefore confusing. These cartoon in the correct size could show where features are in the histogram.)

We have now expanded on the figures with the addition of models to highlight what the different techniques reveal about the N-terminus shape. Peptide cartoons above the NMR data have been modified to scale to make it easier to interpret the data.

Reviewer #4 (Remarks to the Author):

In this manuscript, Murphy et al. showed that a defined architecture within a disordered region contributes to E3 ligase function. The authors discovered that the expected disordered substrate recognition domain of RNF4 might have a defined conformation before SUMO binding using the smFRET and NMR methods. They further mutated the highly basic linker. Using smFRET and molecular dynamics simulations, the authors proposed that electrostatic interactions may contribute to the formation of the highly flexible but compact structure. This manuscript addressed a significant question – what is the mechanism by which SUMO substrate is delivered to the RING - by proposing an electrostatic driven compact structure within a disordered region. Overall, I am impressed by the quality of the paper. However, the proposed mechanism still suffers from multiple deficiencies. My major concerns are that the proposed electrostatic interactions and the global three-dimensional architecture are not well defined in the manuscript. Thus, I think the paper is a bit immature to be accepted as it is. If the authors could address the points discussed below, it could be a very significant and interesting paper.

Major concerns

1. One of the major discoveries of the paper is that “electrostatic interactions involving the highly basic region linking the substrate recognition and RING domains juxtaposed those regions and mediated substrate ubiquitination”. However, it is not clear what residues the basic region interacts with? Is there an acidic region? Or are there any specific acidic residues involved in the electrostatic interactions with the basic region? Are these residues located in any specific SIM region? These questions are important to define the proposed electrostatic interactions.

We agree these points are important. Please see our response to a similar comment from reviewer 3 point 1 (text p8, line 5-p9, line 6; p9, line 7-23; p13, line 23 – p14, line 2).

2. The smFRET focuses on the conformation of the SIM region and demonstrated that the conformation of the SIM region may be primed for SUMO binding. The NMR data, on the other hand, suggested that both termini fold back towards the middle of the peptide. From Figure 3D and Extended data Figure 19, it seems that not only the basic linker, other regions, including residue ~105-115 may also have interactions with other residues. But these interactions are not discussed in the manuscript. Could the interactions involving residues ~105-115 be important for the highly flexible but compact structure? Could the authors use smFRET to measure the conformational change between the basic linker and the RING domain (i.e. the C-terminal region, residues 87-133) and to show whether the conformation of this region is defined or disordered before and after SUMO binding? Or could the mutagenesis experiments demonstrate the roles of residues ~105-115?

On the first point, as outlined in the revised manuscript (Fig. 4A) there is a stretch of acidic residues that follows the arginine-rich motif in the linker (~90-105). Mutations of these residues resulted in changes to the RNF4N peptide according to S.E.C and smFRET, highlighting the role in charge segregation and peptide shape (text p8, line 5-p9, line 7-23). The residues in the region on 110-115

include what was originally thought to be a fifth SIM (V110, Y111, V112) as these residues were sensitive to SUMO titration during NMR analysis (Xu, Y., Plechanovova, A., Simpson, P., Marchant, J., Leidecker, O., Kraatz, S., Hay, R. T., and Matthews, S. J. (2014) Structural insight into SUMO chain recognition and manipulation by the ubiquitin ligase RNF4. *Nature communications* **5**, 4217). Further investigation of these residues however revealed no contribution to SUMO binding. Their absence in the NMR is likely a result of conformational dynamics and further highlights the SIMs/RING domain linker as being a “hinge” between these two domains.

On the second point, our revised manuscript now includes a smFRET analysis of full-length RNF4 measuring the interaction between the SIMs (close to the arginine-rich motif) and RING domain. This analysis revealed direct interaction between the SIMs and RING domain, validating the compact shape of the RNF4 protein. Additionally, Δ Basic mutations also caused a significant reduction in the percentage of molecule in a compact state, further validating our model for the extended conformation with RNF4 Δ Basic having reduced ubiquitination activity (text p10, line 1-21).

3. The authors stated that their studies “have revealed a global three-dimensional architecture within this “disordered” region, with highly flexible but compact structures facilitating productive binding of SUMO substrate.” However, it is not clear how this 3D architecture is defined. Does the entire peptide (residue 32-133) or only SIM+basic linker region (residue 32-88) have a 3D architecture? What characterizes the compact structures?

To address this point a series of models have been incorporated to highlight which peptide region is being focussed on in each figure. Figure 1, outlines the RNF4 domains to scale. Figure 2, reveals the compaction of the SIMs region. Figure 3, reveals compaction of the entire N-terminal peptide. Figure 4, reveals the role of electrostatics in driving peptide compaction. Figure 5, reveals compaction in the full-length RNF4 protein. Figure 7, proposed a structure/function model for the RNF4 compact shape and ligase activity (text p13-15).

4. The molecular dynamics simulations showed the distances between residues 80 and 133 change upon the mutation. However, in Extended data Table 4, the authors did not report distances change between residues 32-79 upon mutation. Are these distances change significantly upon mutation? If not, what could be the reason if the basic residues are critical to maintaining the compact structure in the N-terminal region?

We have now included the distance from the N-terminus to the midpoint for the simulations to Supplementary data Table 4. The distance does increase for the Δ Basic mutant as expected, however the variation across simulations due to the increased conformational flexibility of the extended conformations compared to the more compact configurations limit this analysis. To include more insights into this behaviour we have now incorporated new conformational clustering analysis of the final 100 ns of each trajectory (Figure 4E-F, Supplementary figure 16, text p9, line 7-23).

Minor Concerns

1. The basic residues are mutated to either serine or alanine. Why are they mutated to different residues?

A mixture of serine and alanine was used as alanine alone caused peptide aggregation during single-molecule analysis. This was likely due to the additional hydrophobic residues in proximity to the

hydrophobic SIM residues. This was overcome by have ~50/50 alanine/serine split (text p8, line 14-15).

2. *On Page 18, "Production trajectories were performed for." This sentence is not completed. How long did the production trajectories were performed for?*

We thank the reviewer for noting this typographic error and have corrected it to read 1 μ s (text p21, line 1).

REVIEWER COMMENTS

Reviewer #1 (Remarks to the Author):

The manuscript is considerably improved with the additional data strengthening their conclusions. The schematics also assist with interpretation of the figures/manuscript.

I have no further concerns.

Reviewer #3 (Remarks to the Author):

The manuscript is much improved and easier to follow. However, the key criticism remains that while the biophysical data is abundant, it is less clear what it reveals about the biology of RNF4. My main concern is that the role of the basic region is not as important as the authors are proposing. Some of the data that reinforced this concern in the previous version has simply been removed (Extended data Fig12, extended data table 4). Meanwhile, the new data such as Extended data figure 19 isn't compelling that the basic region is functionally important and critical control data isn't shown (in this case the difference between RNF4 KO and RNF expressing cells).

Regarding the specific points previously raised:

1. On the mechanistic side, there is little explanation or data to explain why the basic residues are important for function: What they are doing to facilitate compaction of the SIMs? What are they interacting with? There seem to be acidic patches in the sequence. Are they interacting with the basic region?

Here it was asked that the functional significance of the basic residues be addressed. The authors provide a new hydropathy plot and one new mutant (Δ acidic). This mutant shows different behaviour on the analytical gel filtration but by smFRET does not look significantly different to WT, in contrast to what the authors state ("modestly increased EFRET", p8). This does not answer why the basic region facilitates compaction. There are simple analyses that could be performed that would

strengthen the case that the basic and acid regions are functionally important. For instance, looking at the sequence conservation between species should be very revealing. If these regions are very important they should be highly conserved both in terms of composition and relative spacing.

2. The Δ basic mutant appears to be important for compaction of the ligase, which brings substrate and RING together. However, there is no mechanism presented here that explains why and how. There is also no data that physically show that the RING domains are actually close to the substrate binding SIMs. Without addressing these questions all we know is that mutating RNF4 perturbs its structure and function.

This comment was intended to ensure that the importance of the basic region in bringing the RING and substrate together was tested. The new data presented in Figure 5 strengthens the authors arguments and shows that RING and SIMs indeed come together and that the Δ basic mutant reduces this behaviour. But what is the expected distance between the 2 dyes for the observed EFRET values? This information seems to be very important. Also, it is interesting that the reduction in the high-FRET states between WT and Δ basic is ~50 % with and without substrate bound. But why is there a general reduction in the high-FRET state when the substrate (tetra-SUMO) is present (32 instead of 52 % for WT and 17 instead of 25 % for Δ basic)? It seems counterintuitive that substrate-binding reduces the ability to deliver the substrate to the RING. This should be discussed in the main text.

3. On the physiological side, some cellular data would help demonstrate that the decrease in activity resulting from mutation of the basic residues is functionally relevant. The authors have previously characterized and measured the physiological functions of RNF4 in cells (Tatham et al. (2008); Yin et al. (2012)). It is therefore surprising that they didn't test in this study whether their in vitro findings are important for RNF4's cellular function.

This was a key point as it asked for some cellular data to demonstrate the physiological importance of the biophysical measurements. It is good that there has been an attempt to do this but disappointing that the assay has not been performed as in the authors previous works. Specifically, a key condition is missing from the data, namely quantification of PML bodies in the RNF4 KO cells. Without this it is not possible to say what contribution RNF4 is making to PML degradation in the experiment nor how much of this contribution the Δ basic mutant is making. If the effect of the Δ basic mutant is physiologically relevant, it should be close to the phenotype of an RNF4 knockout. In addition, a western blot for PML is missing to, which was performed in their original study (Figure 7 in: Tatham, M. H., Geoffroy, M. C., Shen, L., Plechanovova, A., Hattersley, N., Jaffray, E. G., Palvimo, J. J., and Hay, R. T. (2008) RNF4 is a poly-SUMO-specific E3 ubiquitin ligase required for arsenic-induced PML degradation. *Nature Cell Biology* 10, 538-546).

4. SUMO Binding in Extended data Fig 12b looks impaired in the gel shift assay in contrast to what is stated in the main text (p4 l86 "retained SUMO chain binding"). – It seems more correct to replace the original data and explain in the manuscript why it conflicts with other datasets.

5. The table in extended data table 4 suggests that the end-to-end distance is relatively similar between WT and Δ basic. (8.4 vs. 10.1). These values are both within the standard deviation of the measurements. Is there now a significant opening in the Δ basic mutant or not? It is good that new analysis has been added but statements like “The RNF4N Δ Basic mutant displayed a clear reduction in loop and hairpin structures... (Fig. 4F and Supplementary Fig, 16)” on page 9 should be avoided without statistically significant quantification.

6. The NMR experiments with N- or C-terminal spin-labels hardly show an effect. Why are only residues 74 and 75 slightly affected and what about the rest of the construct? – To make the NMR clearer to readers, please provide all spectra in the supplemental part. In addition, it is important to state which residues were lost and could thus not be analysed (in Methods or an additional Table).

7. The manuscript is very short and there is no discussion. How do the results change our view of RNF4 mechanism and function? How does this contrast with our understanding of other ligases? For instance, one interesting comparison might be Cullin-RING E3s and how they achieve substrate ubiquitination (Substrate adaptor on one end, RING on the other, activation by Nedd8 so that the RING can span the distance to the substrate (e.g. Duda et al. (2008) Cell). – The new article format improves the quality of the manuscript. Although the comparison to Cullin-RING-ligases is useful, they are very different to ssRINGS, as the authors also stated. Yet, while Beak et al showed how Neddylation effectively brings RING and substrate together, a clear take-home message from this RNF4 work is lacking. Rather than simply state that the data “highlights how our finding with RNF could offer insight”, please explain what the insight is.

8. Generally this manuscript makes it hard to find information about constructs and more. The Figures are not intuitive and proper labelling would make it much easier to understand them (e.g. the little cartoons above the PRE NMR data have the wrong size and are therefore confusing. These cartoon in the correct size could show where features are in the histogram.) – The revised version of the manuscript has much clearer figures.

Due to the new article format of the revised manuscript, additional issues have appeared in the new text:

1) The second sentence in the abstract “Structural disorder is prevalent within ssE3s particularly in substrate binding and domain linking regions” and third sentence in the second paragraph on page 2 “Structural disorder is high in ssE3s, with about 25% of all residues predicted to reside in disordered regions. These disordered regions are mainly located in the substrate binding domains and inter-domain linkers, with the ordered RING domain responsible for E2 binding (9)” implies that most

ssE3s of the RING type would have a high disorder. This in contrast to the fact that the disorder distribution within E3s is no different to the rest of the human proteome, as shown in the paper the authors reference (Bhowmick, P., Panca, R., Guharoy, M., and Tompa, P. (2013) Functional diversity and structural disorder in the human ubiquitination pathway. *PLoS one* 8, e65443). Structural disorder is thus not prevalent within ssE3s but rather similar to proteins in general. It is true though that there is a subset (17.5 % of ssRINGS or 8.5 % of all E3s) of ssE3s with very high disorder (>50 %) to which RNF4 belongs. This means that RNF4 is not a general representative of ssRINGS. This sentence needs to be altered.

2) The sentence “Molecular dynamics simulations were applied and intramolecular diffusion deemed responsible for closing the distance between bound substrate and E2 (9)” implies that the question of substrate ubiquitination by CBL was solved by the compaction modelling in MD simulations. However, in these simulations, substrate and E2 active site get no closer than 20 Å apart (Bhowmick, P., Panca, R., Guharoy, M., and Tompa, P. (2013) Functional diversity and structural disorder in the human ubiquitination pathway. *PLoS one* 8, e65443). The molecular breathing in MD simulations is therefore not responsible for bringing E2 active site and substrate in range for catalysis. This is also in line with Cullin-RING-ligases. If molecular breathing alone was responsible, there would not be any need for Nedd8 to bring RING, E2 and substrate together (which, as pointed out by the authors, is indeed a requirement: Baek, K., Krist, D. T., Prabu, J. R., Hill, S., Klugel, M., Neumaier, L. M., von Gronau, S., Kleiger, G., and Schulman, B. A. (2020) NEDD8 nucleates a multivalent cullin-RING-UBE2D ubiquitin ligation assembly. *Nature* 578, 461-466). This sentence needs to be altered.

3) In the first sentence on page 9, the authors claim the following: “resulting in the observed compaction in RNF4N Δ Acidic”. This is a strong exaggeration of the observed data. While the analytical gel filtration hints into this direction, the modest increase of EFRET 0.68 ± 0.17 to 0.72 ± 0.17 does not justify this claim.

Reviewer #4 (Remarks to the Author):

The authors have satisfactorily addressed all of my concerns. I would recommend the publication of the manuscript.

Comments in original review in Red

Comments in second review in Black

Our response in Blue

Reviewer #3 (Remarks to the Author):

The manuscript is much improved and easier to follow. However, the key criticism remains that while the biophysical data is abundant, it is less clear what it reveals about the biology of RNF4. My main concern is that the role of the basic region is not as important as the authors are proposing. Our data suggest that the basic region is one component of an multidomain E3 ligase that contributes to the activity of the E3 ligase. Thus, we would not expect its ablation to function as a null mutant. We would expect the activity of a cell line expressing the Δ basic mutant to lie somewhere between that of a null cell line and a cell line expressing the wild type protein. The new data presented in Supplementary Fig 24 indicate that this is indeed the case.

Some of the data that reinforced this concern in the previous version has simply been removed (Extended data Fig12, extended data table 4).

The reason for removing this data is discussed below.

Meanwhile, the new data such as Extended data figure 19 isn't compelling that the basic region is functionally important and critical control data isn't shown (in this case the difference between RNF4 KO and RNF expressing cells).

We have generated a new figure incorporating the additional data requested from the RNF4 KO cell line. This data supports our contention that the basic region contributes to the activity of the E3 ligase (Supplementary Fig 24 D)

1. On the mechanistic side, there is little explanation or data to explain why the basic residues are important for function: What they are doing to facilitate compaction of the SIMs? What are they interacting with? There seem to be acidic patches in the sequence. Are they interacting with the basic region?

1. Here it was asked that the functional significance of the basic residues be addressed. The authors provide a new hydropathy plot and one new mutant (Δ acidic). This mutant shows different behaviour on the analytical gel filtration but by smFRET does not look significantly different to WT, in contrast to what the authors state ("modestly increased EFRET", p8). This does not answer why the basic region facilitates compaction. There are simple analyses that could be performed that would strengthen the case that the basic and acid regions are functionally importance. For instance, looking at the sequence conservation between species should be very revealing. If these regions are very important they should be highly conserved both in terms of composition and relative spacing.

1. The sequence conservation across RNF4 from different species is a good idea. We have performed an analysis of RNF4 sequences showing that in line with their functional importance, charge and spacing is conserved. This analysis is included as Supplementary figure 16 and mentioned on page 8. We certainly agree with the reviewer that the observed difference in EFRET is small but this is expected and follows the same trend as observed in the elution profile. We suspect that the difference between the gel filtration and smFRET data is because the gel filtrations if reporting on the whole peptide, while smFRET is only reporting on the distance between the two dyes, in this case just the basic region to C-

terminus. There are only another arginine and 2 lysine residues outside of 77-88 in the whole RNF4N peptide, while there are two patches of acidic residues at 100-109 (6 acidic residues) and in 32-76 (13 acidic residues). Thus the 3 separate charged regions likely all contribute to the compact structures. This is discussed on page 8/9.

2. The Δ basic mutant appears to be important for compaction of the ligase, which brings substrate and RING together. However, there is no mechanism presented here that explains why and how. There is also no data that physically show that the RING domains are actually close to the substrate binding SIMs. Without addressing these questions all we know is that mutating RNF4 perturbs its structure and function.

2. This comment was intended to ensure that the importance of the basic region in bringing the RING and substrate together was tested. The new data presented in Figure 5 strengthens the authors arguments and shows that RING and SIMs indeed come together and that the Δ basic mutant reduces this behaviour.

2. This was a good suggestion and inclusion of this data strengthens the paper.

2a. But what is the expected distance between the 2 dyes for the observed EFRET values? This information seems to be very important.

2a. Most single-molecule FRET studies report, as we do in our manuscript, relative changes in FRET and inter-dye distances instead of absolute distance values. The reasons for this are: i) relative changes in distances are sufficient to determine the impact of mutations with no need to extract absolute values and ii) extracting absolute distances requires to confirm the exact value of the Förster distance (R_0) for each experimental condition and construct. This would normally be challenging. However, in our experiments we can do this and could provide this data (see table at end of document) but inspection of Fig 2A which shows the potential volume occupied by the dyes and why we would prefer not to include this data. As we are dealing with an intrinsically disordered region we do not have a defined structure on which we can build a model that would allow us map distances between the flexible polySUMO substrate bound to the RNF4 SIMs and the ubiquitin loaded E2 bound to the RING.

2b. Also, it is interesting that the reduction in the high-FRET states between WT and Δ basic is ~50 % with and without substrate bound. But why is there a general reduction in the high-FRET state when the substrate (tetra-SUMO) is present (32 instead of 52 % for WT and 17 instead of 25 % for Δ basic)? It seems counterintuitive that substrate-binding reduces the ability to deliver the substrate to the RING. This should be discussed in the main text.

2b. As requested, this is mentioned in the text (see Results page 11). The trends were the same although only half the number of molecules analysed for the dataset with SUMO (~550 without SUMO, ~300 with SUMO) as we were not trying to directly compare them in this experiment.

3. On the physiological side, some cellular data would help demonstrate that the decrease in activity resulting from mutation of the basic residues is functionally relevant. The authors have previously characterized and measured the physiological functions of RNF4 in cells (Tatham et al. (2008); Yin et al. (2012)). It is therefore surprising that they didn't test in this study whether their in vitro findings are important for RNF4's cellular function.

3. This was a key point as it asked for some cellular data to demonstrate the physiological importance of the biophysical measurements. It is good that there has been an attempt to do this but disappointing that the assay has not been performed as in the authors previous works.

3. We carried out the experiment as was described in Tatham et al 2008, Fig 7A. In this paper we did also include Western blotting of RNF4 KO (siRNA) cells compared to wild type cells (Fig 7B) but the reason we carried out an imaging analysis in our present paper was that it is much more discriminating. This is evident from a comparison of Fig 7 A and B from Tatham et al 2008.

3a. Specifically, a key condition is missing from the data, namely quantification of PML bodies in the RNF4 KO cells. Without this it is not possible to say what contribution RNF4 is making to PML degradation in the experiment nor how much of this contribution the Δ basic mutant is making. If the effect of the Δ basic mutant is physiologically relevant, it should be close to the phenotype of an RNF4 knockout.

3a. We collected the images for the RNF4 KO cells and have now carried out the quantitative analysis of PML bodies. This new data is now incorporated into a new Supplementary Fig 24 and demonstrates that the basic region contributes to the activity of the E3 ligase as the Δ basic mutant has reduced activity when compared to the wild type RNF4 but has a higher level of activity than the RNF4 KO cells. We would not expect cells expressing the Δ basic mutant to have the same phenotype as RNF4 KO cells as all of our biochemical experiments suggest that the Δ basic mutant retains activity but is reduced (not eliminated) when compared to wild type. The RNF4 null cells would be equivalent to leaving out the E3 ligase from the in vitro assay. Thus, the basic region is physiologically relevant as it contributes to the activity of the E3 ligase. We have now made it clear in the text that while the basic region contributes to activity it is not the only thing that determines substrate modification (Results page 13, Discussion page 15). This can be compared to the role of the SIMs in substrate binding (described in our 2014 Nat Comms paper) where mutation of individual SIMs has a negligible effect on activity of the E3 ligase. It is only when multiple SIMs are mutated that we see an effect, but they all play a role in substrate binding.

3b. In addition, a western blot for PML is missing to, which was performed in their original study (Figure 7 in: Tatham, M. H., Geoffroy, M. C., Shen, L., Plechanovova, A., Hattersley, N., Jaffray, E. G., Palvimo, J. J., and Hay, R. T. (2008) RNF4 is a poly-SUMO-specific E3 ubiquitin ligase required for arsenic-induced PML degradation. Nature Cell Biology 10, 538-546).

3b. This was not done as it is clear from Fig 7 of Tatham et al 2008 that looking at PML body numbers is a much more discriminating way of determining activity (compare panels A and B of Fig 7, Tatham et al 2008). In this figure WT cells were compared with siRNA RNF4 knockdown cells where you see a large effect. As indicated above we would not expect to see this with the Δ basic mutant.

4. SUMO Binding in Extended data Fig 12b looks impaired in the gel shift assay in contrast to what is stated in the main text (p4 l86 “retained SUMO chain binding”).

4. It seems more correct to replace the original data and explain in the manuscript why it conflicts with other datasets.

Extended data figure 12 was removed as the electrophoretic mobility shift assay was not a great comparison between WT and basic mutant as the different net charges caused by removal of the basic region resulted in very different gel migrations. Additionally, the point of this result was to demonstrate SUMO chain binding with the Δ Basic mutant, which was clearly shown through the fluorescence polarization binding assay (fig 7 H new manuscript).

5. The table in extended data table 4 suggests that the end-to-end distance is relatively similar between WT and Δ basic. (8.4 vs. 10.1). These values are both within the standard deviation of the measurements. Is there now a significant opening in the Δ basic mutant or not?

5. It is good that new analysis has been added but statements like “The RNF4N Δ Basic mutant displayed a clear reduction in loop and hairpin structures... (Fig. 4F and Supplementary Fig, 16)” on page 9 should be avoided without statistically significant quantification.

5. We agree. The text has been modified accordingly (Results page 10)

6. The NMR experiments with N- or C-terminal spin-labels hardly show an effect. Why are only residues 74 and 75 slightly affected and what about the rest of the construct?

6. To make the NMR clearer to readers, please provide all spectra in the supplemental part. In addition, it is important to state which residues were lost and could thus not be analysed (in Methods or an additional Table).

6. All of the NMR spectra have now been added to the Supplementary material (Suppl Fig 11-15). While residues other than 74 and 75 are affected we were rather conservative with what we quoted a PRE value for. We indicated in Results (page 7) that residues 74 and 75 sit directly adjacent to a basic region that experiences significant line broadening, in which signals were either absent or displayed low signal-to-noise in the HSQC spectra (residues 76-89). Besides residue 82 which displayed PREs, residue 77 could be tentatively assigned but not quantified due to absence of signal in the paramagnetic spectrum. The surrounding residues 76, 78-81 and 83-89 were broadened beyond detection (and thus lack assignment) in the non-paramagnetic spectrum and therefore the data is not available for the PRE even qualitatively (however these would likely be affected).

7. The manuscript is very short and there is no discussion. How do the results change our view of RNF4 mechanism and function? How does this contrast with our understanding of other ligases? For instance, one interesting comparison might be Cullin-RING E3s and how they achieve substrate ubiquitination (Substrate adaptor on one end, RING on the other, activation by Nedd8 so that the RING can span the distance to the substrate (e.g. Duda et al. (2008) Cell).

7. The new article format improves the quality of the manuscript. Although the comparison to Cullin-RING-ligases is useful, they are very different to ssRINGS, as the authors also stated. Yet, while Beak et al showed how Neddylation effectively brings RING and substrate together, a clear take-home message from this RNF4 work is lacking. Rather than simply state that the data “highlights how our finding with RNF could offer insight”, please explain what the insight is.

7. Good point. We have elaborated in the Discussion (page 16)

8. Generally this manuscript makes it hard to find information about constructs and more. The Figures are not intuitive and proper labelling would make it much easier to understand them (e.g. the little cartoons above the PRE NMR data have the wrong size and are therefore confusing. These cartoon in the correct size could show where features are in the histogram.)

8. The revised version of the manuscript has much clearer figures.

8. We agree this has improved the presentation.

Due to the new article format of the revised manuscript, additional issues have appeared in the new text:

1) The second sentence in the abstract “Structural disorder is prevalent within ssE3s particularly in substrate binding and domain linking regions” and third sentence in the second paragraph on page 2 “Structural disorder is high in ssE3s, with about 25% of all residues predicted to reside in disordered regions. These disordered regions are mainly located in the substrate binding domains and inter-domain linkers, with the ordered RING domain responsible for E2 binding (9)” implies that most ssE3s of the RING type would have a high disorder. This in contrast to the fact that the disorder distribution within E3s is no different to the rest of the human proteome, as shown in the paper the authors reference (Bhowmick, P., Pancsa, R., Guharoy, M., and Tompa, P. (2013) Functional diversity and structural disorder in the human ubiquitination pathway. PloS one 8, e65443). Structural disorder is thus not prevalent within ssE3s but rather similar to proteins in general. It is true though that there is a subset (17.5 % of ssRINGS or 8.5 % of all E3s) of ssE3s with very high disorder (>50 %) to which RNF4 belongs. This means that RNF4 is not a general representative of ssRINGS. This sentence needs to be altered.

1. This is a good point and highlights an important characteristic of RNF4. The text of the abstract and introduction (page 2) have been modified accordingly.

2) The sentence “Molecular dynamics simulations were applied and intramolecular diffusion deemed responsible for closing the distance between bound substrate and E2 (9)” implies that the question of substrate ubiquitination by CBL was solved by the compaction modelling in MD simulations. However, in these simulations, substrate and E2 active site get no closer than 20 Å apart (Bhowmick, P., Pancsa, R., Guharoy, M., and Tompa, P. (2013) Functional diversity and structural disorder in the human ubiquitination pathway. PloS one 8, e65443). The molecular breathing in MD simulations is therefore not responsible for bringing E2 active site and substrate in range for catalysis. This is also in line with Cullin-RING-ligases. If molecular breathing alone was responsible, there would not be any need for Nedd8 to bring RING, E2 and substrate together (which, as pointed out by the authors, is indeed a requirement: Baek, K., Krist, D. T., Prabu, J. R., Hill, S., Klugel, M., Neumaier, L. M., von Gronau, S., Kleiger, G., and Schulman, B. A. (2020) NEDD8 nucleates a multivalent cullin-RING-UBE2D ubiquitin ligation assembly. Nature 578, 461-466). This sentence needs to be altered.

2. The text (page 3) has been modified to reflect this point.

3) In the first sentence on page 9, the authors claim the following: “resulting in the observed compaction in RNF4N Δ Acidic”. This is a strong exaggeration of the observed data. While the analytical gel filtration hints into this direction, the modest increase of EFRET 0.68 ± 0.17 to 0.72 ± 0.17 does not justify this claim.

3. The statement (page 9) has been modified to reflect the modest change in EFRET.

Distance measurements

We have extracted the inter-dye distances from the FRET efficiency values obtained for the different constructs and conditions shown in Figure 5 as requested by the reviewer, and also in Figure 4 and

supplementary Figure 18 for comparison. The FRET distances obtained with the associated errors are shown in the table below,

Figure 4D	Low FRET	Distance (Å)	High FRET	Distance (Å)
WT RNF4N	0.34 ± 0.2	67 ± 8	0.68 ± 0.2	53 ± 7
ΔBasic RNF4N	0.32 ± 0.1	68 ± 7	0.53 ± 0.1	59 ± 4
ΔAcidic RNF4N	0.34 ± 0.12	67 ± 6	0.72 ± 0.1	51 ± 6
Figure 5B&C	Low FRET	Distance (Å)	High FRET	Distance (Å)
WT RNF4	0.34 ± 0.1	67 ± 6	0.56 ± 0.2	58 ± 9
ΔBasic RNF4	0.32 ± 0.1	68 ± 6	0.45 ± 0.2	62 ± 7
Fig. S18	Low FRET	Distance (Å)	High FRET	Distance (Å)
WT RNF4 + SUMO	0.32 ± 0.2	68 ± 9	0.68 ± 0.2	53 ± 9
ΔBasic RNF4 + SUMO	0.33 ± 0.2	68 ± 7	0.48 ± 0.3	61 ± 11

*Errors correspond to the standard error of the gaussian centre determined by non-linear least square fitting of the experimental histogram

Extracting absolute distances requires determination of the exact value of the Förster distance (R_0) for each experimental condition and construct. This would normally be challenging. However as shown in the table above, the position of the gaussian centre obtained for the low FRET population is very consistent across all the constructs and experimental conditions and differs by ~ 1 Å. Thus, the position of the low-FRET population acts as an internal reference for changes in R_0 . The remarkable consistency in the position of the low-FRET band suggest that the R_0 value remains similar across the different constructs and conditions and any difference in FRET efficiency observed in the high-FRET state must reflect a conformational change.

REVIEWERS' COMMENTS:

Reviewer #3 (Remarks to the Author):

The authors have provided new information and edited their text in part. Most importantly, the omitted analysis is now available in Supplementary Figure 24. This data significantly weakens the authors claims that the Δ basic RNF4 has a strong physiological effect. The negative control shows that while there is a tiny increase in PML bodies in Δ basic compared to WT, comparison to the RNF4 knockout shows that Δ basic is as significantly different from knockout as is the wildtype, thereby revealing that this mutant is not strongly reducing activity. The region of interest in this paper might be part of bringing the substrate to the ligase domain but its absence results in an enzyme that does the job nearly as good (3 fold difference in biochemical experiments in Fig. 7 and very low effect in cells in Supplementary Fig. 24). It thus remains an open question what drives substrate (SUMO) ubiquitination by RNF4 and thereby enables its physiological function.

The authors have provided new information and edited their text in part. Most importantly, the omitted analysis is now available in Supplementary Figure 24. This data significantly weakens the authors claims that the Δ basic RNF4 has a strong physiological effect. The negative control shows that while there is a tiny increase in PML bodies in Δ basic compared to WT, comparison to the RNF4 knockout shows that Δ basic is as significantly different from knockout as is the wildtype, thereby revealing that this mutant is not strongly reducing activity. The region of interest in this paper might be part of bringing the substrate to the ligase domain but its absence results in an enzyme that does the job nearly as good (3 fold difference in biochemical experiments in Fig. 7 and very low effect in cells in Supplementary Fig. 24). It thus remains an open question what drives substrate (SUMO) ubiquitination by RNF4 and thereby enables its physiological function.

I find the reviewer's statement that "This data significantly weakens the authors claims that the Δ basic RNF4 has a strong physiological effect" rather puzzling as I have gone through our manuscript and revisions and this is not a claim that we have ever made. We indicated that "RNF4 Δ Basic had a significantly reduced ability to clear PML bodies compared with cells expressing RNF4 WT". The reviewer also states that "there is a tiny increase in PML bodies in Δ basic compared to WT, comparison to the RNF4 knockout shows that Δ basic is as significantly different from knockout as is the wildtype". To avoid any confusion, we have modified the manuscript to include the details of the PML body counting in the text (p14). RNF4 WT reduces PML body numbers by 32.8% when compared to the knockout cells ($p=2.4 \times 10^{-20}$), while RNF4 Δ Basic reduces PML body numbers by 24.1% ($p=5.4 \times 10^{-12}$). The difference between WT and Δ Basic is significant ($p=6.2 \times 10^{-4}$) and represents a 25% reduction in the ability of the Δ Basic to reduce PML body numbers when compared to WT.